# An inorganic mineral-based protocell with prebiotic radiation fitness

Shang Dai [1,2,8], Zhenming Xie [1,8], Binqiang Wang[1,8], Rui Ye[3,8], Xinwen Ou [3], Chen Wang[4], Ning Yu[1], Cheng Huang[1], Jie Zhao[1], Chunhui Cai[1], Furong Zhang[1], Damiano Buratto [1,3], Taimoor Khan[1,3], Yan Qiao [5] ✉, Yuejin Hua [1,6,7] ✉, Ruhong Zhou [1,2,3,7] ✉ & Bing Tian [1,7] ✉

Protocell fitness under extreme prebiotic conditions is critical in understanding the origin of life. However, little is known about protocell's survival and fitness under prebiotic radiations. Here we present a radioresistant protocell model based on assembly of two types of coacervate droplets, which are formed through interactions of inorganic polyphosphate (polyP) with divalent metal cation and cationic tripeptide, respectively. Among the coacervate droplets, only the polyP-Mn droplet is radiotolerant and provides strong protection for recruited proteins. The radiosensitive polyP-tripeptide droplet sequestered with both proteins and DNA could be encapsulated inside the polyP-Mn droplet, and form into a compartmentalized protocell. The protocell protects the inner nucleoid-like condensate through efficient reactive oxygen species' scavenging capacity of intracellular nonenzymic antioxidants including Mn-phosphate and Mn-peptide. Our results demonstrate a radioresistant protocell model with redox reaction system in response to ionizing radiation, which might enable the protocell fitness to prebiotic radiation on the primitive Earth preceding the emergence of enzyme-based fitness. This protocell might also provide applications in synthetic biology as bioreactor or drug delivery system.

The appearance of life on the Earth was a result of a series of geochemical events involving the interaction of prebiotic simple inorganic molecules, followed by the formation of biopolymers, such as peptides and polynucleotides, and the emergence of protocells[1,2]. The protocells, also called primitive cells, were the first life-like entities with functions including proto-metabolism, compartmentalization and/or replication, and are therefore of great significance in the research of life origin, synthetic biology and astrobiology[3,4]. To date, there are three major types of protocells known as membrane-free coacervate droplets, lipid vesicles and arising therefrom the hybrid protocell[5–8].

The self-assembled lipid vesicles have been extensively studied as a protocell model to decipher the emergence of compartmentalization and genetic polymer synthesis in the origin of life[8,9], and its fatty acid or phospholipid bilayers are selectively permeable to solutes[9,10]. The coacervate droplet model which relies on spontaneous sequestration and concentration of polyelectrolytes and biomolecules through liquid-liquid phase separation (LLPS) has received increasing attention[11,12]. The coacervate droplet can be used to emulate some of the critical features that are essential to cellular functions such as biomolecular crowding and cellular communication. Recently,

[1]Institute of Biophysics, College of Life Sciences, Zhejiang University, Hangzhou, China. [2]Shanghai Institute for Advanced Study of Zhejiang University, Shanghai, China. [3]School of Physics, Institute of Quantitative Biology, Zhejiang University, Hangzhou, China. [4]College of Pharmaceutical Science, Zhejiang University, Hangzhou, China. [5]Institute of Chemistry, Chinese Academy of Sciences, Beijing, China. [6]Qian Xuesen Collaborative Research Center of Astrochemistry and Space Life Sciences, Ningbo University, Ningbo, China. [7]Cancer Center, Zhejiang University, Hangzhou, China. [8]These authors contributed equally: Shang Dai, Zhenming Xie, Binqiang Wang, Rui Ye. ✉e-mail: yanqiao@iccas.ac.cn; yjhua@zju.edu.cn; rhzhou@zju.edu.cn; tianbing@zju.edu.cn

membrane-bounded hybrid protocells with living material assembly in coacervate microdroplets were developed[6,13]. However, there remains a missing link between prebiotic extreme scenarios such as radiations and adaptive evolution of protocells under radiation stress, if the protocells emerged early. The primordial intensity of terrestrial radioactivity was estimated up to $4 \times 10^3$ time higher than at present[14]. The ionizing radiations from cosmic rays and intrinsic radioisotopes on the early Earth may have a profound influence on the emergence and persistence of life[15]. Theses ionizing radiations including γ-ray drive prebiotic chemistry[14]; on the other hand, they also carry deleterious effects on the biomolecules (RNA, DNA and proteins) of life due to generation of ROS from water radiolysis[15,16]. The radiation-induced oxidative damage even under anoxic conditions on the primitive Earth is not supportive of the protocell formation and evolution because ROS would oxidize biomacromolecules. Unfortunately, there is no study that includes the effects of prebiotic ionizing radiation on protocells.

Moreover, the prevalent building blocks used in the coacervate protocell models are synthetic organic polymers or modern biomacromolecules[17], which might have not existed on the primitive Earth with such extreme conditions. Thus, there remains many challenges to answer the question of the origin and fitness of protocells.

Inorganic polyphosphate (polyP) is a linear negatively-charged polymer consisting of tens to hundreds of orthophosphate (Pi) residues linked by high-energy phosphoanhydride bonds[18]. The polyP appeared on the Earth long before the emergence of biological molecules, and it was a plausible source of energy and phosphate donor for ATP and polynucleotide syntheses in the early evolution of life[18,19]. As a prebiotic inorganic mineral, it is believed that polyP originates from volatile condensates of phosphate rock at elevated temperatures in volcanoes (Fig. 1a), where mineral metal ions including manganese ion ($Mn^{2+}$) existed[20,21]. Among the primitive metal ions, $Mn^{2+}$ in complexes with small molecules provides a beneficial antioxidant activity without the pro-oxidant side effects of other redox active metals[22]. Since the pioneer work by Arthur Kornberg et al. on variety of physiological functions of polyP[18,23], it is proposed that polyP was involved in the origin and survival of life by providing a flexible and polyanionic scaffold to assemble and orientation of biomolecules in prebiotic cells. Today, polyP is ubiquitously present in all living organisms and acts as primordial protein chaperone[24] and cation chelator[25]. It also plays roles in virulence[26] and cell cycle[27]. We have demonstrated that accumulated polyP and thereafter its metabolite (Pi) complexed with cellular $Mn^{2+}$ defends against oxidative stress in the evolutionarily ancient bacterium *Deinococcus radiodurans*[28], which is known for its tolerance to

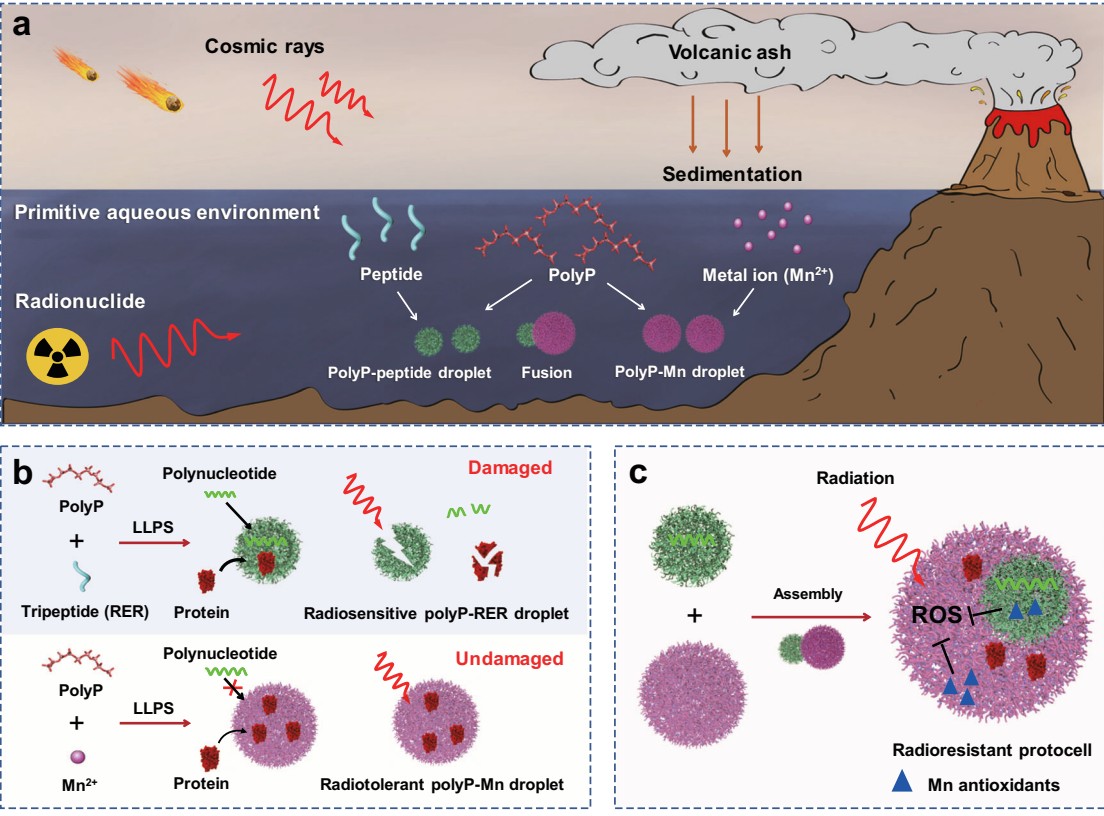

**Fig. 1 | Diagram illustrating the radioresistant protocell model based on phase separation of polyP with simple molecules. a** Prebiotic scenarios for primitive geochemical events leading to the appearance of polyP, Mn and biopolymers on the early Earth[1,14]. Inorganic polyP could be originated from volatile condensates of phosphate rock at elevated temperatures in volcanic eruptions where mineral metal ions[19], including $Mn^{2+}$ deposited in sedimentation[21]. PolyP might interact with simple cationic ligands ($Mn^{2+}$, short peptides) to form LLPS (liquid-liquid phase separation) coacervate droplets and further protocells in the primitive aqueous environment. However, ionizing radiation induced ROS through water radiolysis have deleterious impact on biomacromolecules[16]; **b, c** A radioresistant protocell model (this work). **b** Formation and properties of LLPS coacervates using polyP, $Mn^{2+}$ and tripeptides as the basic building blocks, respectively. PolyP provides the scaffold coordinated with the simple ligands. The polyP-Mn and polyP-RER coacervate droplets demonstrated distinct properties including recruitment ability on biomolecules (polynucleotides and protein) and radiotolerance. Dotted arrows indicate protein recruitment for characterizing the coacervates. The polyP-RER droplets were sensitive to radiation with the results of droplet disassembly and oxidative damages of recruited biomolecules, while the polyP-Mn droplets were radiotolerant; **c** The radioresistant compartmental protocell model enables prebiotic radiation fitness. The protocell assembly through spontaneous fusion of polyP-Mn and polyP-RER droplets in a bottom-up manner. The cytoplasm-like polyP-Mn droplets encapsulated the polyP-RER-polynucleotide as a nucleoid-like condensate inside, and provided protection of the radiosensitive "nucleoid" and recruited proteins against radiation-induced oxidative damage through ROS scavenging by nonenzymic Mn antioxidants, including Mn-RER and Mn-Pi (See details in Fig. 6).

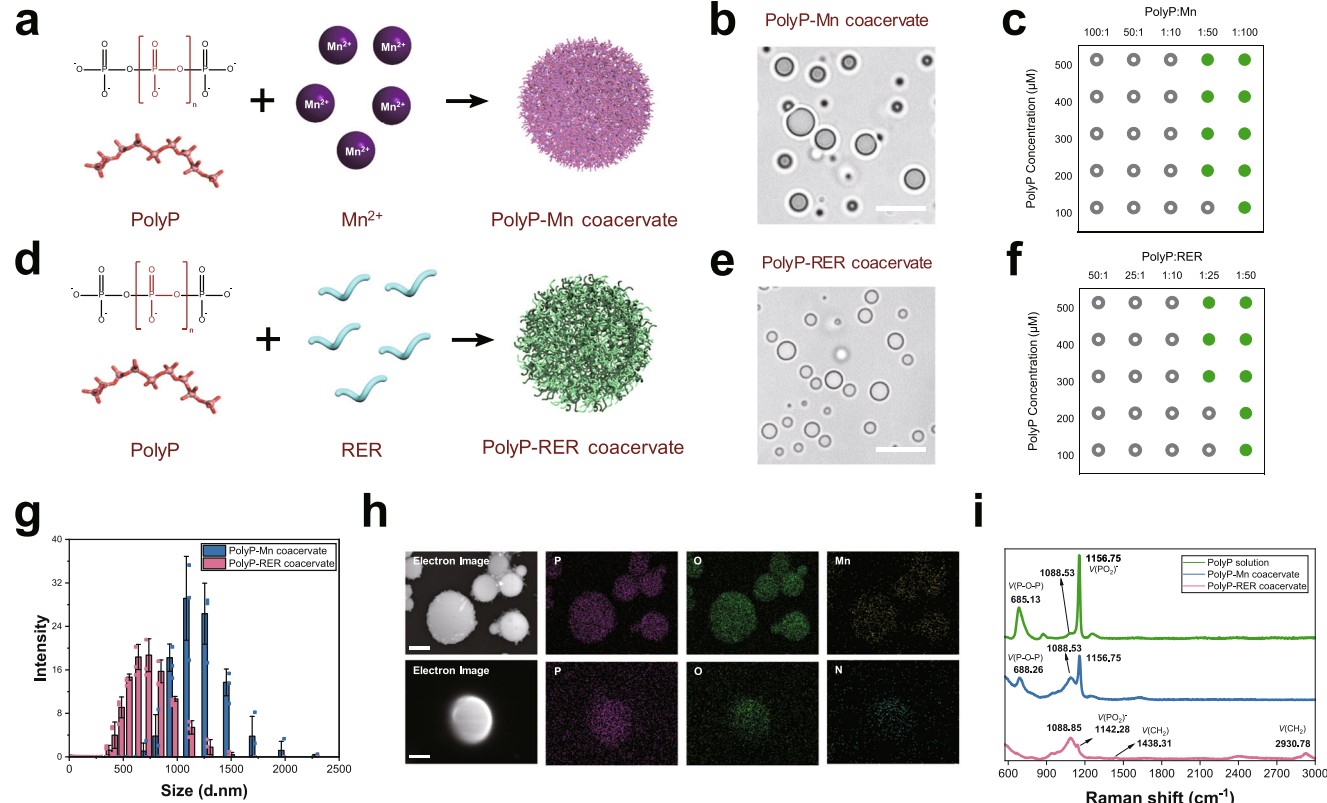

**Fig. 2 | Formation and characterization of coacervate microdroplet through liquid-liquid phase separation of polyP and simple molecules. a** Diagrams of polyP-Mn coacervate microdroplet formation through LLPS of polyanionic polyP with cation ligand $Mn^{2+}$; **b** Bright field image of polyP-Mn coacervates. 0.5 mM polyP (P100) was incubated with equal volume of 50 mM $MnCl_2$. Scale bar, 2 μm; **c** Phase distribution of the polyP-Mn with respect to polyP concentration and the molar ratio of polyP to $Mn^{2+}$. Green spheres indicate that spherical polyP-Mn droplets were observed, while gray circles represent no coacervate droplets were observed; **d** Diagrams of polyP-RER coacervate microdroplet formation through LLPS of polyP with cation ligand tripeptide (RER); **e** Bright field image of polyP-RER coacervates. 0.2 mM polyP (P100) was incubated with equal volume of 20 mM RER. Scale bar, 2 μm; Representative images (**b**, **e**) from *n* = 3 independent experiments; **f** Phase distribution of the polyP-RER with respect to polyP concentration and the molar ratio of polyP to RER. Green spheres indicate that spherical polyP-RER droplets were observed, while gray circles represent no coacervate droplets were observed; **g** The distribution of hydrodynamic diameter of polyP-Mn and polyP-RER coacervate microdroplets measured by DLS at 25 °C, pH 7.0. Data are presented as mean ± SD (*n* = 3 independent experiments); **h** SEM-EDS images of polyP-based coacervate microdroplets. Upper panel: polyP-Mn coacervate microdroplets. Elemental mapping showed that phosphorus (P), oxygen (O) and manganese (Mn) colocalized inside the coacervate microdroplets. Scale bar, 2.5 μm; Bottom panel: polyP-RER coacervate microdroplets. Elemental mapping showed that phosphorus (P), oxygen (O) and nitrogen (N) colocalized inside the coacervate microdroplets. Scale bar, 1 μm. Representative images (**h**) from *n* = 3 independent experiments; **i** Raman spectra of polyP solution, polyP-Mn and polyP-RER coacervate microdroplets. Source data are provided as a Source Data file.

chronical radiations (γ-ray and UV) and survival at a high radiation dose of 10 kGy γ-ray[29]. Moreover, radioresistant cells across the three domains of the tree of life (archaea, bacteria, and eukaryotes) accumulate high level of $Mn^{2+}$ (>2 mM as exemplified by *D. radiodurans*) in the form of high-symmetry Mn-antioxidant complexes with small metabolites (e.g. orthophosphate, amino acid, peptide), which can supplant enzymic antioxidants to protect proteins from ROS damage and govern irradiation survival of cells[30,31]. These studies imply a nonenzymic based fitness mechanism which support cell evolution when exposed to radiation-induced oxidative stress.

In this work, we present a radioresistant coacervate protocell assembled with inorganic mineral and biogenic simple molecules as the basic building blocks and fitness under prebiotic radiation stress. Inspired by the radioresistant cells across the three domains of the tree of life, we envision the protocell with prebiotic radiation fitness and set out to screen and construct a compartmental protocell based on coacervates of inorganic polyP with simple cationic ligands ($Mn^{2+}$, short peptides). The compartmental protocell is formed through spontaneous self-assembly of distinctive polyP-Mn coacervate droplets with polynucleotide-containing polyP-tripeptide (RER) coacervate droplets (Fig. 1b, c). The radiotolerant "cytoplasmic" polyP-Mn provides protection for the recruited proteins and the radiosensitive

"nucleoid" [polyP-RER-polynucleotide] against radiation-induced oxidative damage through the intracellular antioxidants.

## Results

### Construction and characterization of polyP-based coacervate microdroplets

The polyphosphate, which plays indispensable roles in life processes, was proposed to be prebiotically originated in volcanoes where metal ions existed[20,21]. Amino acids and peptides were also hypothesized to be formed in primitive Earth conditions[32,33]. Considering the important role of these ancient and simple molecules in life processes, we screened their coacervate droplet formation ability by mixing the polyP solution with different divalent metal cation ($Mn^{2+}$, $Fe^{2+}$, $Ca^{2+}$ and $Mg^{2+}$) or positively charged oligopeptides at room temperature. The results showed that the polyP (n=100, P100) was able to form coacervate microdroplets with divalent metal cations (Fig. 2a, b and Supplementary Fig. 1a). We screened various positively charged amino acids and short peptides for their potential in coacervate microdroplet formation with polyP, and found that polyP preferred to positively charged dipeptides and tripeptide in solution (Fig. 2d, e, Supplementary Fig. 1b). Among them, the tripeptides (RER, ERR, RRE) have better performance in coacervate microdroplet formation. The arginine-

glutamate-arginine (RER) were selected in our following experiments (Supplementary Note for peptide screening). The coacervate microdroplets formed when the polyP solution was incubated with MnCl$_2$ or RER at proportional molar ratios (P100: Mn$^{2+}$ ≤ 1: 50, i.e. Pi: Mn ≤ 2:1; P100: RER ≤ 1: 25, i.e. Pi: RER ≤ 4:1) (Fig. 2c, f), indicating that saturated coordination of polyP and ligands is necessary. However, extreme high concentration solutions of polyP and divalent metal cations resulted in viscous and gel-like materials (Supplementary Fig. 2). For instance, 4 M polyP and 2 M MnCl$_2$ have been used for preparing glass-like PolyP-Mn solid amorphous material[34]; polyP solution added with 1 M Ca$^{2+}$ (Ca/P molar ratio of 0.5) formed into gel-like biomaterial[35]. Two coacervate microdroplets were selected as protocell chassis in our following experiments: one is the polyP-Mn coacervate microdroplets formed by incubation of 0.5 mM polyP with 50 mM Mn$^{2+}$, and the other is the polyP-RER coacervate microdroplets formed by incubation of 0.2 mM polyP with 20 mM peptides. The spherical coacervate microdroplets with a good optical contrast in the aqueous phase were observed in bright field image (Fig. 2b, e). The average size of polyP-Mn was 1110 ± 12.62 nm and that of polyP-RER was 655.33 ± 14.84 nm, as measured by dynamic light scattering analysis (DLS) (Fig. 2g). The zeta potential values of PolyP-Mn and PolyP-RER were net negative at pH range of 2–8 (Supplementary Fig. 3), indicating that both of the two coacervate microdroplets were well dispersed in aqueous solution.

The coacervate microdroplets were then further characterized using SEM-EDS and Raman spectroscopy. The SEM-EDS images demonstrated the co-localization of elements in coacervate microdroplets of polyP-Mn (P, O and Mn) and polyP-RER (P, O and N), respectively (Fig. 2h). The Raman spectra of the polyP-Mn coacervate microdroplets displayed two intense bands at 1156.75 cm$^{-1}$, 688.26 cm$^{-1}$ and a weak band at 1088.53 cm$^{-1}$ (Fig. 2i), which represented PO$_2^-$ stretching vibration, P–O–P deformation and PO$_3^{2-}$ stretching vibration of polyP[34], respectively. In comparison with polyP solution, the observed relatively decreased intensity of band at 1156.75 cm$^{-1}$ (PO$_2^-$) and 688.26 cm$^{-1}$ (P–O–P) in the polyP-Mn might be attributed to binding effect of Mn ion with the internal PO$_2^-$ of polyP chain. EDTA is a divalent metal chelating agent which could abolish the coordination bond of Mn and polyP thus disintegrate the polyP-Mn droplets. Upon exposure to 20 mM EDTA solution, the polyP-Mn droplets disassembled, while the polyP-RER droplets remained intact (Supplementary Fig. 1c). Similarly, the Raman spectra of the polyP-RER coacervate microdroplets also displayed a band at 1088.85 cm$^{-1}$ (PO$_3^{2-}$) and a decreased peak shift at 1142.28 cm$^{-1}$ (PO$_2^-$), while showed up a peptide signature peaks at 1438.31 cm$^{-1}$ and 2930.78 cm$^{-1}$ (Fig. 2i and Supplementary Fig. 4), indicating that RER might interact with the internal PO$_2^-$ by hydrogen bonds or ionic bonds from NH$_2$ group of arginine. Thus, the linear polyP could mediate the formation of coacervate microdroplets through LLPS, which might be majorly dependent on electrostatic interactions between the polyP and coordinated ligands.

Next, we investigated the effects of external salt (NaCl) on these coacervate microdroplets, respectively. There was no change in the morphology of the polyP-Mn droplets following exposure to 0.6 M NaCl solution, whereas the polyP-RER droplets could be tolerant at less than 0.3 M NaCl (Supplementary Fig. 5), indicating that the polyP-Mn had a strong salt tolerance compared with the polyP-RER. This might be attributed to the weaker internal force of polyP-Na than polyP-Mn, but the stronger internal force of polyP-Na than polyP-RER. These results demonstrated that polyP interacts with simple small molecules and forms LLPS coacervate droplets, which might have distinctive properties and functions.

## Different biophysical properties of coacervate microdroplets

The recruitment of biomolecules by protocells plays a vital role in the formation of compartmentalized cell structures and bioreactors[36]. Here, we characterized the capability of polyP-Mn and polyP-RER

droplets on recruiting protein (mCherry) and polynucleotides (single-stranded DNA [ssDNA] and double stranded DNA [dsDNA] labeled with 6-FAM, see the Method and Material). DAPI was used to identify the polyP component in droplets. Following the adding of mCherry, both of the droplets were able to sequestrate mCherry within 30 s (Fig. 3b and Supplementary Fig. 6a). We used the ssDNA to mimic RNA, which might be a key component in primitive cellular evolution of protocells[36,37]. After incubation of the DNA with the respective coacervates, neither ssDNA nor dsDNA could permeate into polyP-Mn droplets (Fig. 3b and Supplementary Fig. 6b, c). Changing the order of addition (DNA, polyP and Mn$^{2+}$) had no effect on the recruitment of DNA by microdroplets, suggesting that the polyP-Mn microdroplets do not sequestrate DNA (Supplementary Fig. 7). On the other hand, the polyP-RER droplets sequestered ssDNA or dsDNA within a short time (30 s) (Fig. 3b and Supplementary Fig. 6b, c). These results indicated the distinct recruitment properties of polyP-Mn and polyP-RER on biomolecules (polynucleotides and protein) (Fig. 3a). The uptake of DNA by polyP-RER rather than polyP-Mn may be attributed to the lower interaction energy difference between DNA-RER and polyP-RER than that of between DNA-Mn$^{2+}$ and polyP-Mn$^{2+}$ (Fig. 3e), as detailed below using molecular dynamic simulations.

Fluorescence recovery after photobleaching (FRAP) experiments were performed to analyze the LLPS features of the coacervate microdroplets. A single photobleaching pulse led to the stepwise bleaching of the core area. The exposed area in the polyP-Mn-mCherry and the polyP-RER-mCherry showed fluorescence recovery (>60%) as compared to their initial intensity in a short time (60 s) after bleaching, respectively (Fig. 3c and Supplementary Fig. 8). These results confirmed that the liquid-like phases and fluidity of the droplets enables unconstrained protein motion leading to homogeneous distribution of the protein. Moreover, the exposed area in the polyP-RER-ssDNA labeled with FAM showed a fluorescence recovery to approximate 50% in 60 s as compared to the initial intensity after bleaching (Fig. 3d), demonstrating that the liquid-like phases and fluidity of polyP-RER enable the motion and homogeneous distribution of polynucleotides.

Additionally, we explored the potential mechanism behind the distinct recruitment properties of polyP-Mn and polyP-RER droplets on polynucleotides. We hypothesize these physical characteristics of the two types of coacervate microdroplets may be critical to their recruitment properties. Thus, we performed molecular dynamics (MD) simulations to investigate the interactions between ssDNA and the two types of droplets (Fig. 3e–g and Supplementary Figs. 9–11). In the polyP-Mn droplets, the interaction energy of polyP with Mn$^{2+}$, and ssDNA with Mn$^{2+}$ were compared. As shown in Fig. 3e, the average interaction energy of polyP and Mn$^{2+}$ (−1450 kJ mol$^{-1}$) was found much stronger than that of ssDNA with Mn$^{2+}$ ion (−650 kJ mol$^{-1}$), and the binding energy difference was about 800 kJ mol$^{-1}$. Whereas in the polyP-RER droplets, the average interaction energy of polyP with RER peptide (−800 kJ mol$^{-1}$) and ssDNA with RER peptide (−430 kJ mol$^{-1}$) were relatively close, and their binding energy difference (370 kJ mol$^{-1}$) was much smaller than that in polyP-Mn droplets (800 kJ mol$^{-1}$). These results manifested that Mn$^{2+}$ ion has much higher propensity to bind with polyP than ssDNA, which should be unfavorable for ssDNA uptake by the polyP-Mn droplets. On the other hand, the difference in binding propensity of RER peptide to polyP and to ssDNA is much smaller, which should be favorable for ssDNA uptake by the polyP-RER droplets. Importantly, all these findings from simulation were in agreement with our experimental results.

Furthermore, we also performed all-atom molecular dynamics simulations to construct the clusters of polyP and RER, polyP and Mn$^{2+}$ to mimic the polyP-RER and polyP-Mn LLPS droplets (Supplementary Figs. 9 and 10), respectively. We analyzed the electrostatic surface potential of the polyP and Mn$^{2+}$ ions mixed cluster, as well as the polyP and RER mixed cluster. Figure 3f showed that the surface of the polyP and Mn$^{2+}$ mixed cluster was mainly negatively charged, while the

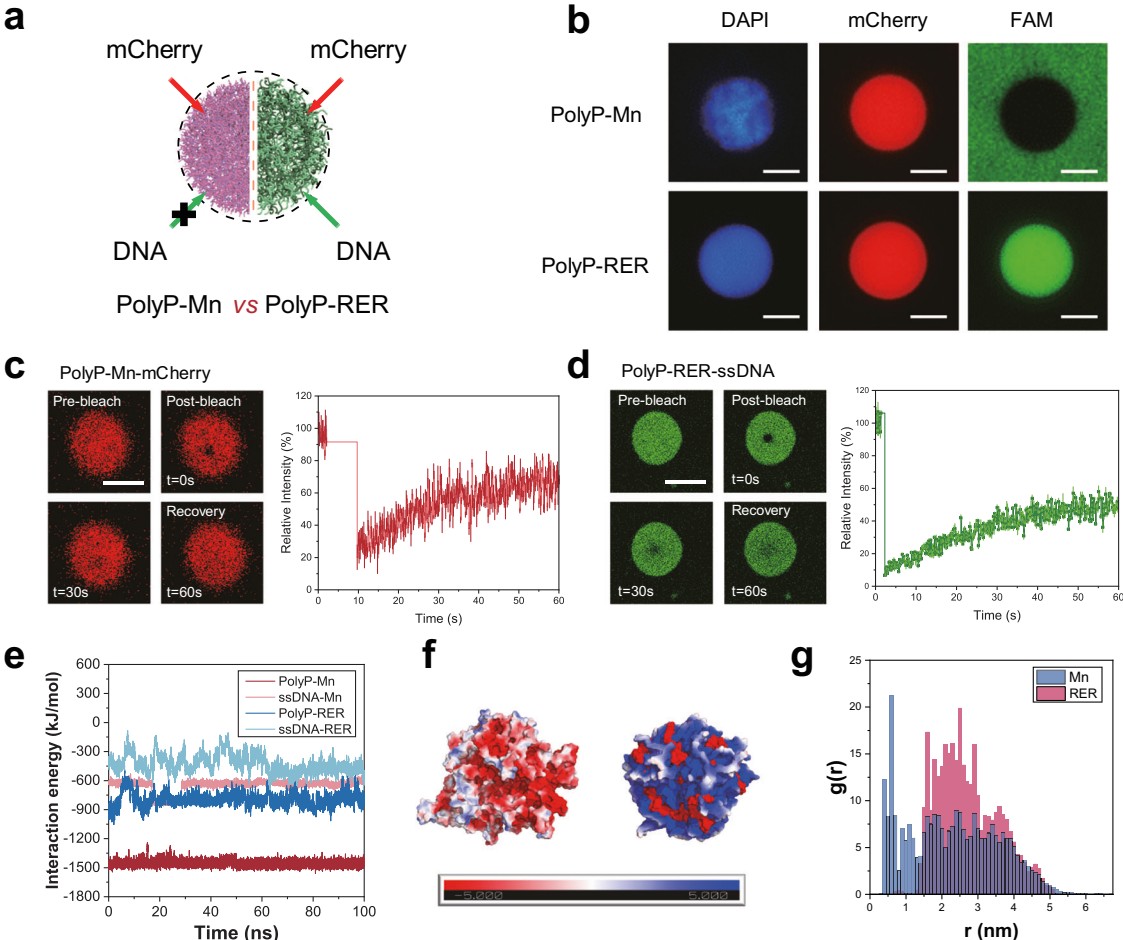

**Fig. 3 | Biophysical properties of polyP-Mn and polyP-RER coacervate microdroplets. a** Scheme of coacervate microdroplets' sequestration on protein (mCherry) and DNA; **b** Fluorescent Microscopy images of recruitment of polyP-Mn and polyP-RER coacervate microdroplets on protein (mCherry, red fluorescence) and ssDNA (FAM, green fluorescent), respectively. PolyP was stained by DAPI (blue fluorescent). Scale bars, 1 μm. Representative images (**b**) from $n = 5$ independent experiments; FRAP assays of polyp-Mn-mCherry (**c**) and polyP-RER-ssDNA coacervate microdroplets (**d**). Fluorescence recovery after photobleaching (left panels) and FRAP recovery curves of the coacervate microdroplets (right panels) were shown. Representative images from $n = 3$ independent experiments. Data are presented as mean ± SD ($n = 3$ independent experiments). **e** The interaction energies (including van der Waals and Coulombic energies) between polyP and $Mn^{2+}$ ion (red line), ssDNA and $Mn^{2+}$ ion (pink line), polyP and RER (blue line), and ssDNA and RER (cyan line). **f** The final surface electrostatic potential of the polyP and $Mn^{2+}$ mixed cluster (left), the polyP and RER mixed cluster (right). Blue, red, and white represent the regions of positive, negative, and neutral surface electrostatic potential, respectively. The color scale represents the range of the surface electrostatic potential, where the minimum (red) and maximum (blue) value is −5 kT/e and 5 kT/e, respectively. **g** Radial distribution functions, g(r), of $Mn^{2+}$ ions (blue bar) and RER (red bar) around the center of the clusters. The results are derived from 100-ns simulations. Source data are provided as a Source Data file.

surface of the polyP and RER mixed cluster was distributed with positively charged RER. The radial distribution functions (RDFs) of $Mn^{2+}$ and RER around the center of the two types of clusters were then calculated to understand the difference of the electrostatic surface potentials. As shown in Fig. 3g, the highest peak of $Mn^{2+}$ ions locate at r = 0.6 nm, while the highest peak of RER located at r = 2.5 nm. This may be due to the fact that $Mn^{2+}$ ions are smaller and more tightly bound to polyP, making it easier to be wrapped in the cluster, while RER peptides are larger and have weaker interactions with polyP and can be dispersedly distributed in the area closer to the surface of the cluster (see details in Supplementary Figs. 9 and 10). Considering the negatively charged ssDNA surface, the ssDNA prefers to approach to and be "pulled" into the polyP-RER through its interaction with RER, rather than to the polyP-Mn with negatively charged surface. These findings indicate that the different electrostatic interaction between DNA and droplets were crucial for the recruitment ability of the droplets on DNA. Moreover, the polyP-Mn were formed by strong coordination bonds, while the internal force among the molecules in polyP-RER cluster were mainly hydrogen bonds and salt bridges as calculated

from molecular dynamics simulation (Supplementary Fig. 11). These findings also suggest that the bonds formed in polyP-RER cluster could be dissociated relatively easier, thus the RER could interact with exogenous molecule such as polynucleotides.

## Radiotolerant polyP-Mn coacervate droplet efficiently protect the recruited proteins

Next, we evaluated the radiotolerant potential of the coacervate droplets, considering that ionizing radiations such as γ-ray influence profoundly on the persistence of life on the early Earth[15]. It was found that the polyP-tripeptide (including polyP-RER) droplet were sensitive following exposure to 500 Gy γ-ray, as shown by the fluorescence disappearance of recruited biomolecules and droplet disassembly (Fig. 4a and Supplementary Fig. 12a). On the contrast, the polyP-Mn, polyP-Ca and polyP-Mg coacervate droplets were still intact following exposure to 500 Gy γ-ray (Supplementary Fig. 12b). After exposure to γ-ray, polyP-Fe coacervate droplets aggregated and formed into precipitation, which might be due to oxidation of $Fe^{2+}$ caused by γ-ray induced ROS. Radiation did not significantly promote the degradation

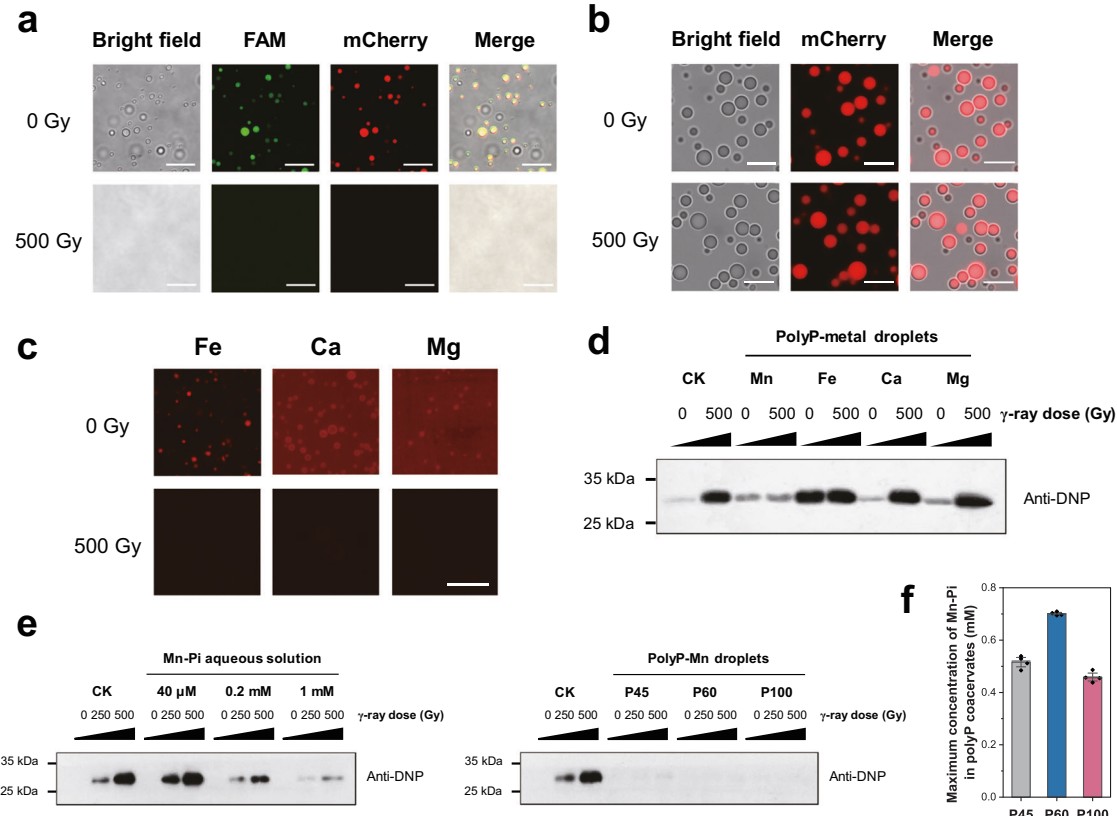

**Fig. 4 | The radiotolerant polyP-Mn coacervate droplet efficiently protected the recruited proteins. a** Microscope images of polyP-RER coacervate droplets containing FAM-ssDNA (green) and mCherry (red) following exposure to γ-ray. Scale bar, 10 μm; **b** Microscope images of polyP-Mn coacervate droplets containing mCherry (red) following exposure to γ-ray. Scale bar, 10 μm. **c** Microscope images of polyP-metal coacervate droplets containing mCherry (red) following exposure to γ-ray. Scale bar, 10 μm; Representative images (**a**–**c**) from $n = 3$ independent experiments; **d** Western blot assay of protein carbonylation of mCherry in solution

(CK) and polyP-metal coacervates, respectively. Anti-DNP, anti- 2,4-dinitrophenyl. **e** Western blot assay of protein carbonylation of mCherry in solution containing Mn-Pi at different concentrations (left panel) or in polyP-Mn droplets containing polyP of different length (P45, P60 and P100) (right panel). Experiments were independently repeated three times with similar results (**d**, **e**). **f** Molybdenum blue assay of free phosphate (Pi) release from polyP-Mn droplets. Data are presented as mean ± SD ($n = 4$ independent experiments). Uncropped blots and source data are provided as a Source Data file.

of polyP as shown by the in vitro assays of polyP solution using Urea-PAGE, molybdenum blue staining as well as $^{31}$P NMR assays (Supplementary Fig. 13), indicating that the polyP scaffold was maintained for the integrity of droplet under γ-ray irradiation. Furthermore, we tested the effects of γ-ray on the mCherry recruited in different polyP-metal coacervate droplets (Fig. 4c and Supplementary Fig. 14). After exposure to 500 Gy γ-ray, fluorescence of mCherry disappeared in the droplets, except of the polyP-Mn coacervate droplets (Fig. 4b, c). Protein carbonylation level of mCherry in the polyP-Mn droplet was remarkably lower than that of in the other polyP-metal droplets (polyP-Fe, polyP-Ca, and polyP-Mg) (Fig. 4d). These suggested that polyP-Mn droplets were radiotolerant and protective to the recruited proteins. The protective effect of polyP-Mn droplets might be attributed to the Mn-antioxidant chemistry. Mn-Pi complex (Mn-HPO$_4$) catalytically scavenges $O_2^{\cdot-}$ $(2O_2^{\cdot-} + 2H^+ \rightarrow H_2O_2 + O_2)$ and were substitutable for SOD[22,31,38,39]. We compared the protection ability of the polyP-Mn droplet on proteins with that of Mn-Pi solution (Fig. 4e). As the concentration of Mn-Pi increased, the carbonylation level of mCherry in the solution decreased after exposure to γ-ray. The polyP-Mn droplets formed with different length of polyPs ($n = 45, 60, 100$) showed lower protein carbonylation level compared with that in the 1 mM Mn-Pi solution under γ-ray radiation (Fig. 4e, Supplementary Fig. 15). A small fraction of released phosphate (Pi) in the polyP-Mn was found and thus might formed into Mn-Pi with excessive Mn$^{2+}$ in the droplets (Fig. 4f), which is consistent with the previous report of slow self-degradation of polyP in solution[35]. Due to the excessive Mn$^{2+}$ in the droplet, the

content of free Pi could be considered as theoretical maximum concentration of the Mn-Pi. We calculated the theoretical maximum concentration of Mn-Pi in the polyP-Mn droplet. For the polyP ($n = 45, 60, 100$)-Mn droplets, the maximum concentration of the Mn-Pi is 0.5 mM, 0.7 mM and 0.5 mM, respectively (Fig. 4f), which are lower than that in the Mn-Pi solution system (1 mM). However, these polyP-Mn droplets displayed stronger protection for proteins than that of Mn-Pi solution. These results suggested that the Mn-antioxidant accumulated in the coacervate droplet could possess more efficient scavenging ROS ability to protect proteins than Mn-antioxidant in solution.

## Hierarchical self-assembly of polyP-Mn[polyP-RER-DNA] protocell

Microdroplet fusion has been considered as a feasible way of protocell assembly and growth[40]. We observed the quick fusion processes among the polyP-Mn droplets or the polyP-RER droplets within 10 s, respectively (Supplementary Fig. 16a). These findings indicated the fluidity and assembly feature of these studied coacervate microdroplets. Moreover, we conducted a time course analysis of the fusion progress of polyP-RER-ssDNA and polyP-RER-mCherry droplets using fluorescent microscopy, respectively. The results showed that two droplets could quickly combine into a single droplet (Supplementary Fig. 16b).

The spontaneous fusion properties of the coacervate microdroplets encouraged us to explore the assembly of a spatially partitioned protocell. Based on different properties of the polyP-Mn and

polyP-RER coacervate microdroplets, we attempted to construct a compartmental protocell containing nucleoid-like condensates through hierarchical self-assembly process (Fig. 5a).

First, we obtained the polyP-RER coacervate droplets with sequestered ssDNA (FAM labeled). As mentioned above, the polyP-RER droplets were able to recruit ssDNA, while the polyP-Mn droplets were not (Fig. 3a). Second, the polyP-RER-ssDNA and polyP-Mn droplets were mixed at equal volumes in a tube. The polyP-Mn droplets were found to encapsulate the polyP-RER-ssDNA through spontaneous rapid fusion within 4 s (Fig. 5c, Supplementary Movie 1), which formed a ssDNA-containing nucleoid-like condensate inside the cytoplasm-like polyP-Mn (Fig. 5b). This result indicated the successful formation of a compartmentalized protocell referred as polyP-Mn[polyP-RER-DNA] in a bottom-up assembly manner. Furthermore, red-fluorescent mCherry was added into the protocell solution for characterizing the spatial structure of protocell and the location of nucleoid-like protoorganelle (Fig. 5d). After incubation, the mCherry was recruited into the protocell and further sequestrated by the "nucleoid" (yellow fluorescence) (Fig. 5e). The mCherry distributed evenly in the whole protocell, while the ssDNA was only located in the "nucleoid" (Fig. 5f, g). In this work, the ssDNA was used to mimic primitive polynucleotides of the RNA world. The compartmentalization of primitive hereditary substance within a protocell is an essential step in the origin of life and a promising route to organization and assembly functional protocells. Membrane-less organelles driven by LLPS, which are ubiquitous in modern cells such as nucleolus[41], play key roles in subcellular spatial organization and cell functions. PolyP "granules" have been found in all three domains of the tree of living cells, and was proposed as a membrane-less organelle formed by LLPS and tied to functions such as cell cycle and cellular protein localization[27,41,42], implying that polyP-based coacervate might be an ancient and important component on the progress of cell origin and evolution. The polyP "granules" organized the nucleoid in bacteria such as *Pseudomonas aeruginosa*, perhaps actively promoting remodel and compaction of the nucleoid[27]. Here we demonstrated the LLPS-driven polyP-Mn coacervate droplet which could encapsulate the DNA-containing polyP-RER through droplet fusion and might provide protection of the "nucleoid" against ionizing radiation-induced damage. Thus, the formation of polyP LLPS-based membrane-free coacervates in equilibrium systems might be relevant to cell evolution on the early Earth and still working in modern cells.

## The polyP LLPS-based protocells protect intracellular "nucleoid" from ionizing radiation damage by non-enzymic antioxidants

Radiolysis of water ($H_2O \rightarrow HO^{\cdot} + H^+ + e_{aq}^-$) induced by γ-ray could produces highly reactive and biologically damaging ROS, including hydroxyl radical ($HO^{\cdot}$) and superoxide anion radical ($O_2^{\cdot-}$), which constitute the most damaging impact of ionizing radiation on cells[16]. The charged $O_2^{\cdot-}$ formed through radiolytic reduction of oxygen molecules by hydrated electron ($e_{aq}^- + O_2 \rightarrow O_2^{\cdot-}$) oxidizes proteins[31], while the $HO^{\cdot}$ majorly cause DNA damages.

We explored the effects of γ-ray radiation ($^{137}Cs$) on the polyP-Mn[polyP-RER-DNA] protocell with DNA inside the nucleoid-like condensate under different radiation doses. Figure 4a showed that the polyP-RER droplets which recruited protein and DNA was completely collapsed under γ-ray radiation. Following the exposure to γ-ray radiation (500 Gy and 1 kGy), the protocell still maintained as spherical droplets with intact "nucleoid" (Fig. 6b) and fluorescence of DNA and mCherry indicated that the radioresistant protocell may provide protection for the DNA and proteins inside the "nucleoid". Raman spectra of the protocell also indicated that the major cytoskeleton component polyP was not substantially damaged following the exposure to γ-ray radiation (<2 kGy) (Fig. 6c).

ROS produced by radiolysis often cause serious DNA damage[15]. We quantitatively evaluated the possible polynucleotide damages inside the protocells by measure the damage of single-stranded DNA (ssDNA, 40 nt) or double-stranded DNA (dsDNA, 40 bp) recruited in the polyP-Mn[polyP-RER-DNA] protocell and the polyP-RER droplet, respectively. The protocell and the polyP-RER droplets were irradiated at doses of 0, 50, 250, 500 and 1 kGy, then subjected to PAGE analysis (Fig. 6d, e). After the irradiation, the damage (degradation) percentage of ssDNA in aqueous solution increased with the irradiation dose, and reaching up to 100% at 1 kGy (Fig. 6e). However, ssDNA in the irradiated protocells showed less damage than that of in aqueous solution and polyP-RER droplet under irradiation at various dose (50–1000 Gy) (Fig. 6d, e). Following exposure to 1 kGy γ-ray, ssDNA damage in the protocells (25.92%) was substantial lower than the DNA degradation level in the polyP-RER-DNA coacervate (67.98%) and the control (100%). Also, dsDNA in the protocells degraded by 39.54%, which was substantially lower than the DNA degradation level in the polyP-RER-DNA coacervate (69.03%) and the control (99.74%) (Fig. 6f, g), indicating that the protocells provided a profound protection on polynucleotide inside the "nucleoid" against radiation stress.

ROS-induced protein damage (carbonylation) usually leads to enzyme inactivation, thus affect physiological function and is lethal to living organisms[15]. We determined the impacts of radiation on enzymes inside the polyP-Mn[polyP-RER-DNA] protocell using the following experiments. First, the protocells recruited with β-galactosidase (β-GAL) were prepared (Supplementary Fig. 17), and the β-galactosidase aqueous solution without any treatment was used as a control. The samples of aqueous solution and droplets containing β-galactosidase were then exposed to γ-ray irradiation at different doses (0–1 kGy). Herein, the increased protein carbonylation levels of the β-GAL was directly proportional to the increase of irradiation dose. However, the protein carbonylation level inside the protocell was remarkably lower than that of in the enzyme solution and the polyP-RER droplet (Fig. 6h). This is consistent with that the polyP-RER droplet without the encapsulation by polyP-Mn could be disrupted by γ-ray irradiation (Fig. 4a). Moreover, the β-GAL activity exhibited a reduction with different degrees in aqueous solution, polyP-RER droplet and protocells under radiation, respectively (Fig. 6i, j). However, the β-GAL inside the protocells still maintain activity and exhibit a significant higher activity than that of in the polyP-RER droplet and the aqueous solution after exposure to irradiation (500 and 1000 Gy) (Fig. 6i, j). Therefore, the protection of peptides (RER) and proteins might be attributed to a potential redox reaction system in the protocells, which provided fitness of the protocell under irradiation.

Furthermore, we investigated the fitness mechanism of the protocells under radiation. First, ROS scavenging ability of the protocell were measured. The protocell showed a strong $O_2^{\cdot-}$ scavenging ability (SOD-like) before and after suffering γ-ray radiation (Fig. 6k). By using the Electron paramagnetic resonance (EPR) with the spin trapping reagent DMPO added in the samples at pre-irradiation for ·OH detection, we performed the ·OH scavenging assay. After irradiation, characteristic signals of DMPO-·OH in different samples (control, polyP solution, polyP-RER coacervate and protocell) were detected (Fig. 6l). The intensity (peak amplitude) of DMPO-·OH in the protocell exhibited a sharply decrease as compared to that of in the other samples, indicating that the protocell has strong ·OH scavenging ability. These results suggested that the protocell has an efficient ROS scavenging system which protected biomolecules from oxidative damage.

$Mn^{2+}$ forms complexes with small molecules (orthophosphate, amino acid, and peptide) and act as non-enzymic antioxidants, e.g., Mn-Pi and Mn-peptide complex catalytically scavenges $O_2^{\cdot-}$ and are substitutable for SOD[22,31,38,39]. In the radioresistant bacterium *D. radiodurans*, polyP chelates free $Mn^{2+}$ and facilitated the accumulation of Mn ions as a "Mn pool" in the cell, and the polyP degradation product (Pi) complexed with Mn scavenged $O_2^{\cdot-}$ and protect proteins in

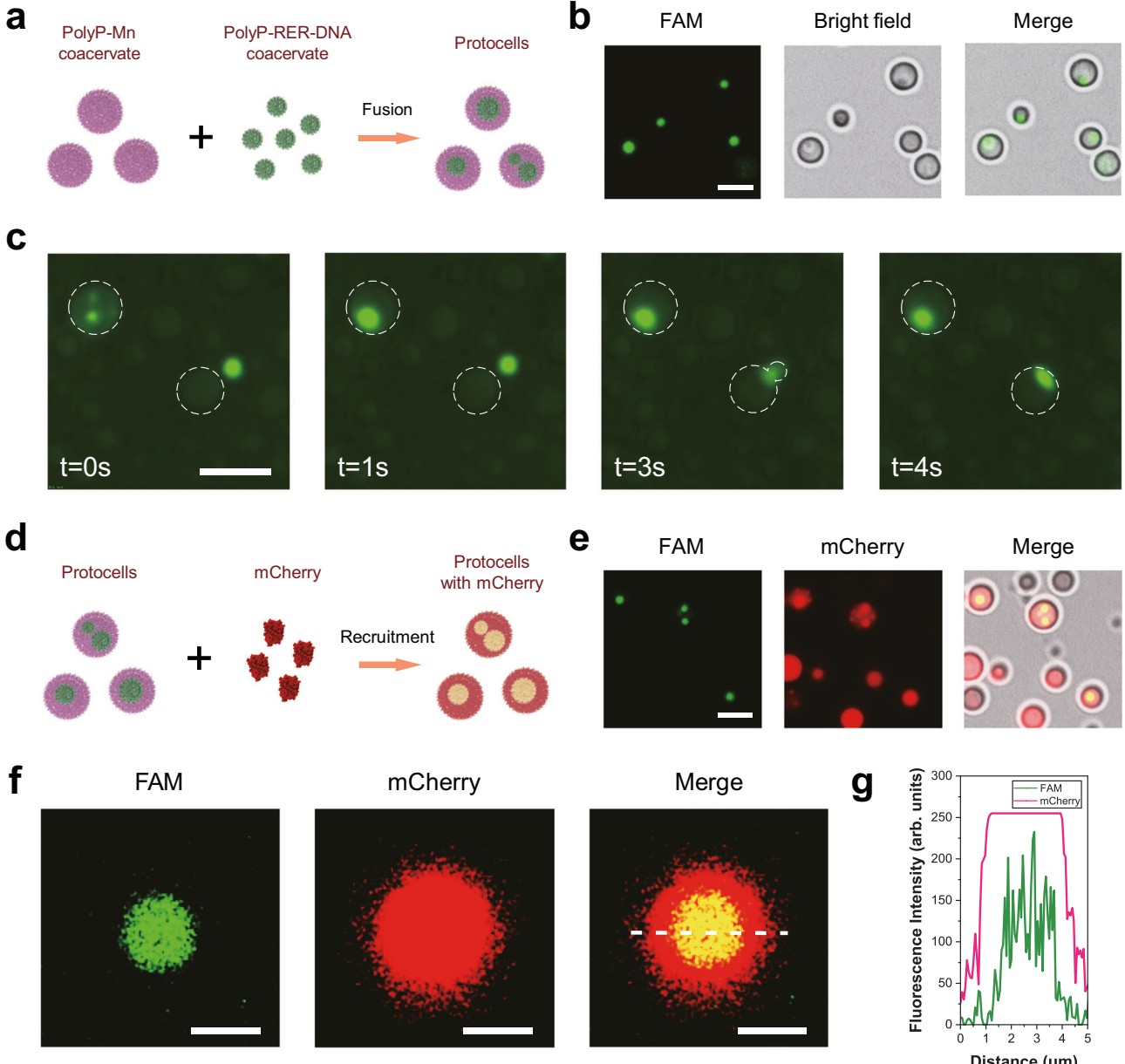

**Fig. 5 | Hierarchical self-assembly of the compartmental protocell polyP-Mn[polyP-RER-DNA]. a** Scheme of protocell assembly. PolyP-Mn droplets encapsulated polyP-RER-DNA droplet by spontaneous fusion and formed a compartmentalized protocell containing nucleoid-like condensates [polyP-RER-DNA]. **b** Microscopy images of protocells with internalized polyP-RER-DNA (green). Scale bars, 2 μm; **c** Time course of spontaneous fusion of polyP-Mn droplets with polyP-RER-DNA droplets. White dotted circle indicated the polyP-Mn droplets. Scale bars, 5 μm; **d** Scheme of sequestration of red-fluorescent mCherry by the protocell for characterizing the protocell's spatial structure. The mCherry structure template for illustration was acquired from RCSB PDB: 4ZIO; **e** Microscopy images of the compartmental protocells with internalized polyP-RER-DNA (green) and mCherry (red). Scale bars, 2 μm; Cross-sectional fluorescence microscopy image (**f**) and corresponding fluorescence intensity profile (dashed line in the Merge image) of a single protocell containing FAM-ssDNA and mCherry (**g**). Scale bars in **f**, 1 μm. The arb. units in **g** indicate arbitrary units. Images in **b**, **c**, **e** and **f** represented the results of three independent experiments. Source data are provided as a Source Data file.

response to radiation[28]. Thus, we investigated the non-enzymic antioxidants in the protocell. First, we identified the presence of Mn complexes in the protocells using ESI-MS assay. The mass spectra analysis revealed the presence of signals of [Mn(RER-H)+Na]⁺ ($m/z = 536.16$) corresponding to the Mn complexes with RER (Mn-RER), and [Mn(HPO₄)+H]⁺ ($m/z = 152.11$) that possibly corresponds to Mn-Pi (Supplementary Fig. 18). Thus, we measured the free $Mn^{2+}$, RER and Pi in the protocells. By EPR assays, we identified the presence of free $Mn^{2+}$ in the protocells exposed to γ-ray at 1 kGy, which exhibited a characteristic six-line EPR signal associated with $Mn^{2+}$ (Supplementary Fig. 19a). We also measured the free Pi in the protocol compared with polyP solution using relative Pi release assays. A small fraction of total

Pi in the polyP solution and the protocell were released with the increase of time, respectively (Supplementary Fig. 19b). The MS spectra analysis also detected [RER+H]²⁺ ($m/z = 230.64$) (Supplementary Fig. 18a), which corresponds to free RER in the protocells. Furthermore, we conducted experiments to verify the ROS scavenging capacity of these Mn complexes in vitro, which were prepared by mixing MnCl₂, Pi, and RER in solution according to the concentration ratio in the protocells (Supplementary Table 1). We verified that the the Mn-RER and Mn-Pi exhibit highly $O_2^-$ scavenging activity (>90%) (Fig. 6k), and Mn-RER-Pi also displayed $O_2^-$ scavenging activities (Supplementary Fig. 20a). EPR assays on DMPO-˙OH indicated that the Mn-RER had the highest ˙OH scavenging activity (Fig. 6l), while the Mn-

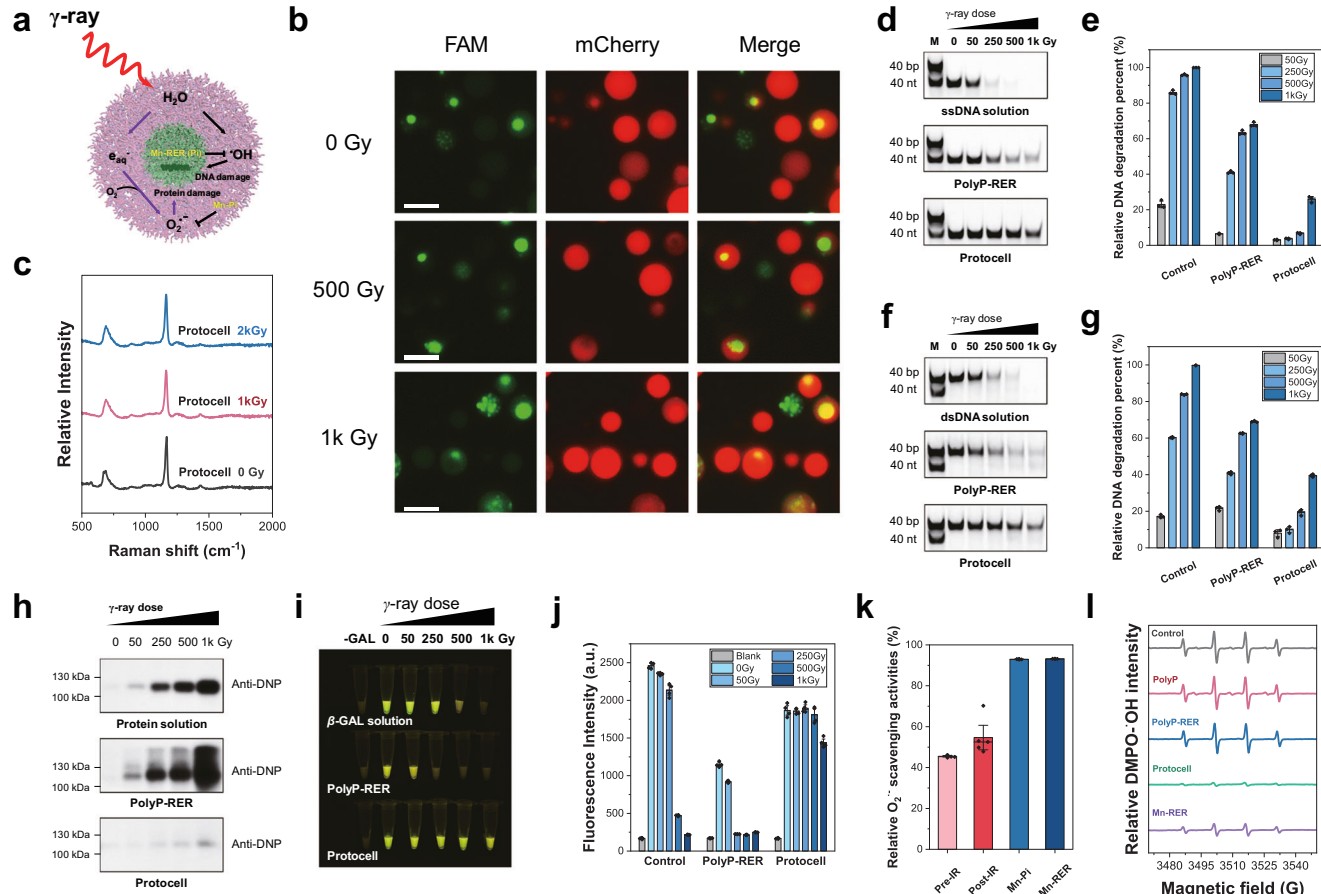

**Fig. 6 | Protection of nucleoid-like condensate and biomacromolecules by small-molecule Mn antioxidants in the protocell after exposed to γ-ray radiation. a** Scheme of the protocell model with nonenzymic redox reactions in response to radiation-induced oxidative stress. Mn-Pi and Mn-RER (Pi), Mn antioxidants complexed with Pi and/or RER. **b** Fluorescent microscope images of the protocells containing FAM-ssDNA (green) and mCherry (red) exposed to γ-ray at different doses (0, 500 and 1 kGy). Scale bar: 5 μm. Images in **b** represented the results of three independent experiments; **c** Raman spectra of polyP in the protocells exposed to γ-ray at 0, 1k and 2 kGy; PAGE analysis of ssDNA damage in the protocell and polyp-RER coacervate exposed to γ-ray at doses of 0-1 kGy (**d**), and corresponding quantitation of relative DNA damage level represented as DNA degradation percentage (**e**). M, DNA marker using synthetic dsDNA (40 bp) and ssDNA (40 nt). A ssDNA solution was used as the control. PolyP-RER, polyP-RER-ssDNA coacervate; Data presented as mean ± SD (*n* = 3 independent experiments). PAGE analysis of dsDNA damage in the protocell and polyp-RER coacervate exposed to γ-ray at doses of 0-1 kGy (**f**), and corresponding quantitation of relative DNA damage level represented as DNA degradation percentage (**g**). M, DNA marker using synthetic dsDNA (40 bp) and ssDNA (40 nt). A dsDNA solution was used as the control. PolyP-RER, polyP-RER-dsDNA coacervate; Data presented as mean ± SD (*n* = 3 independent experiments). Experiments were independently repeated three times with similar results (**d**, **f**). **h** Western blot assay of protein carbonylation of β-GAL solution, polyP-RER coacervates and protocells containing β-GAL and ssDNA exposed to γ-ray at doses of 0-1k Gy, respectively. Experiments were independently repeated three times with similar results. Fluorescent images of β-galactosidase (β-GAL) activity in β-GAL solution, polyP-RER coacervates and protocells containing β-GAL exposed to γ-ray at doses of 0-1 kGy (**i**), and corresponding fluorescence intensity of cleavage products by β-GAL (**j**). -GAL, Blank; Data presented as mean ± SD (*n* = 4 independent experiments). **k** Relatively O$_2^{--}$ scavenging activity of the protocells at pre and post-γ ray radiation (1 kGy) as well as the scavenging activity of Mn-Pi and Mn-RER in solution. Deionized water was considered as the control. Data presented as mean ± SD (*n*=5 independent experiments for Pre-IR and Post-IR samples, *n* = 4 independent experiments for Mn-Pi and Mn-RER samples); **l** EPR spectra of DMPO-•OH intensity of polyP solution, polyP-RER coacervates, protocells and Mn-RER exposed to γ-ray at 1 kGy. Deionized water was considered as the control. PolyP-RER, polyP-RER-ssDNA coacervate. Uncropped gels, blots and source data are provided as a Source Data file.

Pi did not (Supplementary Fig. 20b). Mn-RER complex may come from coordination of Mn$^{2+}$ in the "cytoplasm" with RER in the "nucleoid" of the protocell. Unlike in aqueous solution, the internal molecules in liquid-liquid phase separation system has the characteristics of both condense and isolation[43,44], thus the Mn-antioxidants could be accumulated in the protocell. These results suggest that the antioxidant Mn complexes could be formed by Mn$^{2+}$ with free small molecules (Pi, RER) and contribute to radioresistance of the protocell.

## Discussion

This work presents a protocell model based on coacervates of inorganic polyP with simple cationic molecules, supporting a prebiotic and radioresistant mechanism in which proteins and spatially partitioned polynucleotides can be protected by nonenzymatic Mn-small

molecule antioxidant complexes against radiation-induced ROS damage (Fig. 6). The inorganic linear polyP provides the scaffold coordinated with biogenic simple ligands (Mn$^{2+}$ and tripeptide) required for assembly of the compartmental protocell. The Mn levels in primary well-preserved carbonate rock from the Archean Eon (4 Ga to 2.5 Ga) were proposed to be up to 1% Mn[45]. And the Mn$^{2+}$ was inferred to have been released substantially by the high-temperature hydrothermal solutions during the Early Archean and stored in anoxic seawater in dissolved state[46]. Glutamic acids (Glu) have been identified in meteorites, which might deliver amino acids to the early Earth[47]. Peptides could be formed in the presence of minerals, which was a possible prebiotic process. It was found that Glu binds to the hydroxylapatite, a mineral of composition Ca$_5$(OH)(PO$_4$)$_3$, and the basic amino acid arginine (Arg) binds to the clay minerals such as kaolinite

for further polymerization by condensing agent[47]. Moreover, the tripeptide glutathione was suggested as a model prebiotic tripeptide to stabilize iron–sulfur clusters[48].

Our findings provide insights into the important pathway towards the organization of protoorganelles and proto-metabolic systems on the early Earth. Also, the hierarchical assembly of polyP-simple molecule droplets (polyP-Mn and polyP-RER) appears as a promising route for further organization and assembly of multifunctional artificial cells. The fusion of two droplets occurs to reduce the total interfacial energy. The way that two coacervate droplets fuse (i.e., 1 in 2, or 2 in 1) is largely dependent on how the interfacial energy can be minimized[49,50]. The phase separation between the two droplets is driven by complete wetting in multiphase condensate organization, which is generally due to relative interfacial tensions at play in the system containing multiple phases with different compositions[50,51]. The phase separation between the two droplets occurs to minimize the total interfacial energy[49]. Although the laboratory study on protocell may not reflect the real pathways that lead to the origin of cells on the early Earth, it is proving to be invaluable in reconstructing plausible pathways and prebiotic scenarios to depict the origin of life.

This protocell model with fitness under radiation stress could be a candidate for cell evolution from molecular evolution. Radiation induced $O_2^{\cdot-}$ and $\cdot OH$ could be efficiently scavenged by Mn-Pi and Mn-RER(Pi), respectively. Particularly, peptides and proteins entrapped in the irradiated protocells were profoundly protected by the nonenzymatic Mn antioxidants against $O_2^{\cdot-}$, which is important to the fitness of protocells because the charged $O_2^{\cdot-}$ in irradiated cells selectively damaging proteins[30]. Thus, the polyP LLPS-based protocell with redox reaction system might be operative in the earliest cells in response to prebiotic radiation stress before the emergence of enzyme-based fitness on the early Earth. Beside the application in prebiotic research, particularly in understanding the impact of radiation on the protocells, the facile compartmental protocell can also be applied in synthetic biology as bioreactor, and/or in drug delivery of nucleic acid and proteins.

# Methods

## Materials
Sodium polyphosphate (P100, average polymerization length is 100, units) was kindly provided by professor Adolfo Saiardi, University College London, UK. Sodium polyphosphate (P60 and P45, average polymerization length is 60 and 45 units) was kindly provided by professor Toshikazu Shiba, Kitasato University, Japan. Oligopeptides (RR, KK, HH, RRR, KKK, HHH, RER, RRE, ERR), 6-Carboxyfluorescein (6-FAM) labeled single strand DNA (5′- GGTCAGTTGA-CATTTTTCTTATCGGCGCTCTACCATCCGT-3′) and non-labeled downstream oligonucleotide (5′- ACGGATGGTAGAGCGCCGATAA-GAAAAAT GTCAACTGACC-3′) were synthesized by Sangon Co. (Shanghai, China). Manganese chloride tetrahydrate, ethylenediaminetetraacetic acid disodium salt (EDTA-Na$_2$), sodium chloride, 4′,6-diamidino-2-phenylindole (DAPI), ammonium molybdate tetrahydrate, sodium sulfite anhydrous, hydroquinone, glycerol and Tris-HCl solution were purchased from Sangon Co. (Shanghai, China). The protein carbonyl assay kit used for protein oxidative damage analysis was purchased from Abcam Co. (Shanghai, China). $\beta$-galactosidase was purchased from Solarbi Co. (Beijing, China). Fluorescein di-$\beta$-D-galactopyranoside (FDG) used as the substrate of $\beta$-galactosidase was purchased from MedChemExpress Co. (MCE, Shanghai, China). The free radical trapping agent 5,5-Dimethyl-1-pyrroline N-oxide (DMPO) was purchased from Aladdin Co. (Shanghai, China). All reagents were of analytical grade or the highest purity available and Milli Q purified water (18.2 MΩ·cm) was used in all the experiments.

Fluorescence protein mCherry was purified following the previous method[27]. Briefly, the *mCherry* gene on pET28a was expressed in *Escherichia coli* BL21, and purified using the His-Trap HP column (GE

Healthcare, USA) and eluted with elution buffer B (20 mM Tris-HCl [pH 7.5], 500 mM NaCl, and 500 mM imidazole). The collected protein fractions were then purified on a Superdex 200 10/300 GL column (GE Healthcare) with elution buffer C (20 mM Tris-HCl [pH 7.5] and 250 mM NaCl). Fractions containing the target proteins were pooled, concentrated, flash frozen in liquid nitrogen, and stored in −80 °C.

## Preparation of coacervate microdroplets
The coacervate microdroplets dispersions were prepared by mixing aqueous solution of anionic polyP (P100) and cationic metal salt/oligopeptides at room temperature as follows.

*Preparation of polyP-Mn coacervate microdroplets*: The polyP-Mn coacervate microdroplets were prepared by mixing aqueous solution of P100 and MnCl$_2$ at different molar ratio. Typically, identical volumes of 0.5 mM polyP ($n = 100$) solution were mixed with 50 mM MnCl$_2$ solution at room temperature.

*Preparation of polyP-RER coacervate microdroplets:* The polyP-RER coacervate microdroplets were prepared by mixing aqueous solution of P100 and RER at different molar ratio. Typically, identical volumes of 0.2 mM polyP ($n = 100$) solution were mixed with 20 mM RER solution at room temperature.

## Fluorescence microscopy
Fluorescence microscopy was performed using a Nikon ECLIPSE Ti2-U Inverted Fluorescence Microscope (Nikon Co., Japan). The bright field and fluorescent images of coacervate microdroplets were obtained under the fluorescence microscope. FAM-DNA (green fluorescence) was detected at 488 nm excitation and 500–530 nm emission wavelengths; mCherry (red fluorescence) was detected at 514 nm excitation and 585–635 nm emission wavelengths. The images were analyzed by NIS-Elements D software (v4.60.00).

## Confocal laser scanning microscopy
Confocal laser scanning microscopy (CLSM) was performed using a Leica THUNDER Imager Live Cell confocal laser scanning microscope. The bright field and fluorescent images of coacervate microdroplets were obtained under the confocal microscope. FAM-DNA (green fluorescence) was detected at 488 nm excitation and 500–530 nm emission wavelengths; mCherry (red fluorescence) was detected at 514 nm excitation and 585–635 nm emission wavelengths. To quantitatively analyze the fluorescence signals of 6-FAM and mCherry inside the coacervate microdroplets and protocells, the fluorescence intensity was measured using Leica Application Suite X (LAS X) software (v3.3.0.16799) (Leica Co. Germany).

## Dynamic light scattering analysis
Dynamic light scattering (DLS) analyses were performed using a Malvern Zetasizer Nano-ZS instrument equipped with a 633 nm laser[52]. Samples were injected into a disposable zeta cuvette at 25 °C. The polyP-Mn coacervate microdroplets used for Dynamic light scattering analysis was prepared by mixing identical volumes of 0.2 mM polyP ($n = 100$) with 10 mM MnCl$_2$.

## SEM-EDS
Scanning electron microscope (SEM) analyses were undertaken on Hitachi UHR FE-SEM SU8010 SEM (Hitachi, Japan). The coacervate microdroplets sample was dripped on the copper mesh and dehydrated. The dehydrated samples were coated with gold-palladium and observed under the SEM. Energy-dispersive X-ray spectroscopy (EDS) was used to carry out the elemental distribution analysis.

## Raman spectra
2.5 µL of coacervate droplet sample was dripped on the detection chip (Hooke Instruments, Changchun, China). Then the samples were analyzed using a HOOKE P300 confocal Raman spectrometer (Hooke

Instruments, Changchun, China). For each sample, the acquisition time was set to 10 s and the laser power at the sample was 5.0 mW[53]. All Raman spectra were processed with background subtraction, and baseline correction using HOOKE INTP software (v 1.0) (Hooke Instruments, Changchun, China).

## Recruitment of biomolecules by coacervate microdroplets

To evaluate the sequestration of DNA and protein by coacervate microdroplets, the freshly prepared coacervate microdroplets were incubated with FAM-DNA (green fluorescent) and mCherry protein (red fluorescent), respectively. Briefly, 10 μL coacervate microdroplets was incubated with 2 μL DAPI solution (5 μg/mL), 2 μL FAM-DNA solution (10 μM), and 2 μL mCherry solution (0.5 mg/mL) at room temperature for 5 min. PolyP was stained by blue-fluorescent DAPI. Then, the samples were imaged under Nikon ECLIPSE Ti2-U Inverted Fluorescence Microscope without further treatment. The images were analyzed by NIS-Elements D software (v4.60.00).

## Fluorescence recovery after photobleaching (FRAP)

FRAP was performed on a Leica THUNDER Imager Live Cell confocal laser scanning microscope with a 100× oil objectives. FAM-DNA in coacervate microdroplets was excited with an argon laser at 488 nm, and the emission signals were collected in the range 500–530 nm. The mCherry was excited at 514 nm, and emission signals were collected in the range 585–635 nm. Ten images at attenuated laser intensity (20% intensity) were taken before photobleaching. Photobleaching was performed using a single pulse of argon laser (405 nm) at 20% intensity for FAM-DNA (5 frames), 60% intensity for mCherry (50 frames) through a round region of interest (ROI) of nominal diameter of 1.0–3.0 μm. The laser was then switched back to attenuated intensity and the recovery images were recorded for 74 s (500 frames). The time interval of each frame is 0.148 s. The FRAP data were analyzed using Leica Application Suite X (LAS X) software (v3.3.0.16799).

## Molecular dynamics simulations

We first constructed a polyP chain containing 30 phosphate (Pi) residues. The initial structures of the single-stranded DNA (ssDNA) consisting of 17 bases (5′-GCCGATAAGAAAAATGT-3′) and the 3-amino acid peptide (sequence: RER) were prepared using the PyMOL software[54]. To investigate the interactions between ssDNA and RER, polyP and RER, ssDNA and $Mn^{2+}$ ion, as well as polyP and $Mn^{2+}$ ion, we constructed four systems containing 1 ssDNA and 1 RER peptide, 1 polyP and 1 RER peptide, 1 ssDNA and 1 $Mn^{2+}$ ion, 1 polyP and 1 $Mn^{2+}$ ion, respectively. These systems were solvated in water boxes (with the size of 7 nm × 7 nm × 7 nm) and neutralized with sodium ions, and then followed by a steepest descent minimization and a 100 ns NPT equilibration (300 K and 1 atm). After the equilibration of the simulation systems, a 100 ns production run in the NPT (300 K and 1 atm) ensemble was performed for each system.

Furthermore, we constructed the cluster of polyP and RER peptides, and polyP and $Mn^{2+}$ ions to mimic the LLPS droplets. To facilitate the cluster formation, 10 polyP and 150 RER peptides were randomly placed in a small box (with the size of 8.5 nm × 8.5 nm × 8.5 nm) with a species ratio of 2:1 for phosphate and RER. The small cubic box was then placed in the center of a large box (with the size of 14 nm × 14 nm × 14 nm). For comparison, we also constructed a system with 68 polyP molecules and 1020 $Mn^{2+}$ ions. The systems were solvated in water and neutralized with sodium ions, and then followed by a steepest descent minimization and equilibrated for 100 ns under NPT ensemble (300 K and 1 atm). After the equilibration of the simulation systems, a production run of 400 ns in the NPT (300 K and 1 atm) ensemble was carried out for each system. Then, we analyzed the electrostatic surface potential for two systems using the adaptive Poisson-Boltzmann solver (APBS) plugin in PyMOL software[54]. To compare the interactions of ssDNA with these two systems, we also

calculated the electrostatic surface potential of ssDNA by using a similar procedure. To understand the structural basis of the electrostatic surface potentials, the radial distribution functions (RDFs), g(r), of $Mn^{2+}$ ions and RER peptides with respect to the center of the two types of clusters were calculated. More specifically, the central phosphorus atom of the innermost polyP in the cluster is taken as the reference for the RDF calculation. For the RER peptides, the CB atom of the glutamic acid (E) is used to represent the molecular center.

The CHARMM36 forcefield[55] was used for ssDNA, RER, and sodium ion. The force field parameters of polyP and $Mn^{2+}$ ions were obtained from previous works[56,57]. The TIP3P model was chosen for water molecules. The periodic boundary conditions (PBC) were applied in all three dimensions. The long-range electrostatic interactions were computed with the particle mesh Ewald (PME) method[58]. The short-range electrostatic interactions and the vdW interactions were truncated with a cutoff distance of 1.2 nm. The LINCS algorithm was adopted to constrain the bond vibrations involving hydrogen atoms, allowing for a time step of 2 fs. The system temperature (T = 300 K) was controlled using the velocity-rescaled Berendsen thermostat. The pressure was set to 1 atm using an isotropic Parrinello–Rahman pressostat. All the molecular dynamics simulations were conducted with the GROMACS package (version 2020.6)[59]. Snapshots were rendered by the PyMOL software (v2.5)[54] and ChimeraX software (v1.4)[60].

## Preparation of polyP-Mn[polyP-RER-DNA] coacervate protocells

For preparation of the polyP-Mn[polyP-RER-ssDNA] coacervate protocells, 5 μL the polyP-RER microdroplets were used to sequester 2.5 μL 4 μM DNA (FAM labeled) to form polyP-RER-DNA as described above. Then, 5 μL polyP-Mn coacervate microdroplets were mixed with 7.5 μL polyP-RER-DNA coacervate microdroplets. After incubating the mixture for spontaneous fusion at room temperature, the polyP-Mn[polyP-RER-DNA] coacervate protocells were obtained. Red-fluorescent mCherry was used for characterizing the protocell's spatial structure.

## γ-ray radiation treatment

For radiation treatments, $^{137}$Cs was used as the γ-ray radiation source (facility of the Institute of Crops and Nuclear Technology Utilization, Zhejiang Academy of Agricultural Sciences, Zhejiang, China). The samples were irradiated at room temperature for 2 h with γ-ray at different dose rate. Different dose rates were achieved by adjusting the distance between samples and the γ-ray source, and doses were measured using the potassium dichromate/silver ($K_2Cr_2O_7$/Ag) method[61] at the assistance of the staff at the facility.

## DNA damage assay

DNA damage in the protocell was measured using polyacrylamide gel electrophoretic analysis (PAGE). To measure multiple DNA damages, we used double-stranded DNA (dsDNA, 40 bp) or ssDNA (40 nt) inside the polyP-Mn[polyP-RER-DNA] protocell and polyP-RER-DNA coacervate. The dsDNA fragments are obtained by annealing single-stranded DNA with FAM-labeled and complementary DNA single-stranded without FAM-labeled. The protocell samples were prepared as described above. Then, the protocell and the nucleoid-like polyP-RER-DNA coacervate were treated with irradiation at doses of 0 Gy, 50 Gy, 250 Gy, 500 Gy and 1 kGy. Then, 15 μL samples were lysed by 5 μL glycerol loading buffer solution (33% glycerol, 1×TBE buffer, 250 mM EDTA, 10 mM NaOH), then samples were subjected to the PAGE analysis. PAGE analysis was carried out in 1×TBE buffer at 150 V (constant voltage) for 30 min. The gels were visualized using Typhoon FLA 9500 imaging system (GE health, America), and the gel images were analyzed by ImageJ software (v1.51j8). Quantitation of relative DNA damage level was represented as intensity percentage of the DNA standard without irradiation. The untreated DNA solution sample

prepared by mixing 2.5 µL of 4 µM FAM-DNA in 12.5 µL deionized water was considered as control, and the DNA in solution exposed to γ-ray radiation were used for comparison purpose. Data was presented as mean ± SD, $n = 3$.

## Protein oxidation assay

$β$-Galactosidase was used to evaluate the impacts of radiation on proteins inside the polyP-Mn[polyP-RER-DNA] protocell. The protocell and polyP-RER-DNA coacervate droplet with sequestered $β$-galactosidase were prepared in 20 µL Tris-HCl solution (100 mM, pH 6.8) by mixing 8 µL of the protocell or the coacervate with 2 µL $β$-galactosidase solution (0.7 mg/mL). The samples were placed under γ-ray irradiation (0–1000 Gy) from a $^{137}$Cs radiation source. The enzyme solution was used as control.

**β-Galactosidase enzyme activity assay.** $β$-Galactosidase enzyme activity assay was performed following the method as described previously[62]. After irradiation, the samples were lysed by 4 µL EDTA solution (100 mM, pH 10). 5 µL NaCl solution (1 M) and 9 µL deionized water were added into the sample solution to make up to 48 µL. After vortex for 2 min, 2 µL nonfluorescent substrate FDG solution (150 µmol) was added into the mixture. Samples in the absence of $β$-galactosidase were used as negative controls (Blank). The substrate FDG will release a fluorescein molecule after being degraded by $β$-galactosidase. After 2 h of reaction at room temperature, 25 µL of the mixed solution was taken out to measure the fluorescence value at Ex460/Em515 with a microplate reader. The measured fluorescence intensity was used to reflect the enzymatic activity of $β$-galactosidase. Data presented as mean ± SD, $n = 4$.

**Western blot assay of protein carbonylation.** The irradiated samples were lysed by 10 µL EDTA solution (100 mM, pH 10). Measurement of protein carbonylation was conducted by using Protein Carbonyl Assay Kit (ab178020, Abcam Co. Shanghai, China). Briefly, the carbonyl groups in protein side chains are derivatized to 2,4-dinitrophenylhydrazone (DNP-hydrazone) by reaction with 2,4-dinitrophenylhydrazine (DNPH). The DNP moieties are detected by western blotting using an anti-DNP antibody. From the results of western blot, we can identify the levels of oxidative damage in the samples. Uncropped and unprocessed scans of the blots were provided in the Source Data file.

## O$_2^{•-}$ scavenging activities assay

Scavenging activities of protocells on $O_2^{•-}$ scavenging were measured using the Total Superoxide Dismutase Assay Kit with WST-8 (Beyotime Co, China). In brief, WST-8 can react with $O_2^{•-}$ to produce water-soluble Formazan dye, and this reaction step can be inhibited by SOD-like scavengers, so their $O_2^{•-}$ scavenging activity is negatively correlated with the production amount of dye. The relative scavenging activities of $O_2^{•-}$ can be calculated by colorimetric analysis.

The protocells at pre-irradiation condition and 2 h of post-irradiation condition (1 kGy γ-ray) were harvested by centrifuge at 12,000 rpm for 2 min for measuring their $O_2^{•-}$ scavenging activities. The centrifuged protocell precipitate was lysed with 5 µL 100 mM EDTA and diluted to 30 µL with deionized water. A 20 µL of the lysate was used for subsequent determination of superoxide anion scavenging activity, the deionized water was used as a control. For the $O_2^{•-}$ scavenging activity assay of Mn complex, Mn complexes with small molecules were prepared as shown in the following Supplementary Table 1. The absorbance of the reaction product was measured at 450 nm. Relative $O_2^{•-}$ scavenging percentage = (A$_{control}$ − A$_{sample}$)/(A$_{control}$ − A$_{blank}$) × 100%. Data presented as mean ± SD ($n = 4$).

## EPR spectroscopy measurement and analysis

EPR spectroscopy measurement of hydroxyl radical were performed following the method as described previously[63]. The samples used for EPR analysis were the polyP solution, polyP-RER-DNA coacervates, polyP-Mn[polyP-RER-DNA] protocell and Mn complexes. Deionized water was used as control. The polyP coacervates and protocell samples were prepared as described above. The final concentration of ssDNA was 0.5 µM. For the scavenging activity assay of Mn complex, Mn complexes with RER or Pi were prepared as shown in the Supplementary Table 1. All samples were added with free radical trapping agent (DMPO, final concentration 200 mM). When all the samples were ready for next step, the prepared samples were treated with 1 kGy γ-ray irradiation. After the irradiation, the samples stored on dry ice were subjected to subsequent EPR measurement: 100 µL of sample solution was transferred into a capillary glass tube, and was in turn introduced into a quartz tube. The recording of the EPR spectrum of the sample was initiated at room temperature with an X-band EPR spectrometer (Bruker ESRA-300, Germany). The instrument conditions were the following: frequency of approximately 9.8 GHz with 100 kHz modulation, central field ± sweep width of 3514G ± 60 G, microwave power of 20 mW, time constant of 81.92 ms, sweep time 40 s. The obtained EPR spectra were analyzed using Bruker WinEPR Processing software (v2.22Rev.12) (Bruker Co.).

## Statistics analysis

Statistical parameters, including the definitions and exact values of $n$ (number of experiments) and deviations are reported in the figures and corresponding figure legends. Data are expressed as mean ± SD. Statistical analyses were performed using the OriginPro software (V9.0.0; OriginLab Corporation, MA, USA).

## Data availability

All data supporting the findings of this work can be found in the Source Data file and Supplementary Information. The data generated in this study are provided in the Source Data file. Source data have been deposited in the Figshare database (Figshare https://doi.org/10.6084/m9.figshare.24116619). Source data are provided with this paper.

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

## Acknowledgements

We thank professor Adolfo Saiardi from University College London and Toshikazu Shiba from Kitasato University for providing the polyphosphates. We thank the staff at the facility of the Institute of Crops and Nuclear Technology Utilization, Zhejiang Academy of Agricultural Sciences for their assistance with radiation treatment. This work was financially supported by the grant from the National Key R&D Program of China (2019YFA0905400 to B.T.; 2021YFA1201200 to R.Z.), the grants from the National Natural Science Foundation of China (Grant 32170028 to B.T.; U1967217 to R.Z.; 32200020 to S.D.), the Starry Night Science Fund of Zhejiang University Shanghai Institute for Advanced Study (SN-ZJU-SIAS-003 to R.Z.), the National Center of Technology Innovation for Biopharmaceuticals (NCTIB2022HS02010 to R.Z.), the National Independent Innovation Demonstration Zone Shanghai Zhangjiang Major Projects (ZJZX2020014 to R.Z.).

## Author contributions

B.T. and S.D. were responsible for the experimental design. S.D., Z.X., B.W., C.W., N.Y, C.H., J.Z., C.C. and F.Z. performed the experiments. R.Y., R.Z., X.O., D.B and T.K. performed the molecular dynamics simulations and analysis. B.T., R.Z., S.D. Y.H. and Y.Q. performed data analysis and drafted the manuscript. All authors read and approved the version to be published.

## Competing interests

The authors declare no competing interests.
