## [Peer Review File · Nature Communications]

REVIEWER COMMENTS

Reviewer #1 (Remarks to the Author):

The authors investigate the protection afforded by coacervates on nucleic acids and proteins. The coacervates were either made from polyphosphate and peptide (RER) or polyphosphate and Mn²⁺. The Mn²⁺ containing coacervates protected against damage from radiation whereas the peptide containing coacervates were not as effective. Surprisingly, Mn²⁺ containing coacervates did not uptake DNA but could uptake protein. Peptide containing coacervates could uptake both. This distinction is important since the authors show that only Mn²⁺ containing coacervates protect against radiation. However, since both types of coacervates could fuse with each other, the resulting structure (which the authors call a protocell) could protect DNA.

The work is unique and of clear interest to the origins field. Protecting protocells from reactive oxygen species through the activity of prebiotically plausible components that function somewhat similarly to superoxide dismutase is certainly striking. The authors used a wide variety of techniques, from spectroscopy to computational chemistry and biochemistry.

I have only minor criticisms.

How was radiation administered? Doses of radiation are indicated, but how this was done was not clear. Clarifying this is important so that others can build on this work.

Did the authors try to add DNA and polyphosphate simultaneously to Mn²⁺ or change the order of addition to determine whether they could get encapsulated DNA?

Mn²⁺, Fe²⁺, Ca²⁺ and Mg²⁺ were tested. Did the authors try Zn²⁺, another metal that has been invoked in origins theories?

"however, its fatty acid or phospholipid bilayers are highly impermeable to several solutes due to lack of transporters." It's fair to make this distinction with coacervates, but the permeability problem is frequently over emphasized. Fatty acid vesicles are quite permeable (doi: 10.1073/pnas.0408440102 & DOI:10.1038/nature07018) and the selective permeability of fatty acid membranes could be advantageous.

Protocells could have emerged relatively early or late with respect to the appearance of the building blocks of life. This manuscript assumes that protocells emerged early, which is fine, but the authors should note that this is only one possibility. Perhaps a later emergence of protocells could have found milder conditions?

"The primordial intensity of terrestrial radioactivity was estimated up to 4×10^3 time higher at around 4.56 Ga than at present." This is similar to the point I was making above. People usually don't argue that life emerged at 4.56 Ga.

A comment on the prebiotic plausibility of Arg and Glu and on short peptides would be good, since RER is necessary for the described chemistry. Incidentally, tripeptide complexes with metals have been invoked before in origins scenarios (DOI: 10.1038/nchem.2817).

A comment on the prebiotic concentrations of Mn²⁺ would be helpful.

"Amino acids and peptides were also proved to be formed in primitive Earth conditions." While I agree with the sentiment, "prove" is a bit too strong here.

Reviewer #2 (Remarks to the Author):

This manuscript demonstrated the construction of a protocell model based on coacervation of inorganic polyphosphate with divalent metal cation (Mn^{2+}) and cationic tripeptide, respectively. The biophysical properties of coacervate microdroplets has been characterized that shows the polyP-Mn droplets were radiotolerant and provides strong protection for recruited proteins, while the polyP-tripeptide droplets were not. By mixing the two populations, it was found that polyP-RER droplets fused into PolyP-Mn droplets to form polyP-RER in polyP-Mn multi-compartmentalised structure. DNA or proteins entrapped inside polyP-RER phase were efficiently protected by the polyP-Mn layer under γ -ray radiation, that was claim by nonenzymatic Mn-small molecule antioxidant complexes against radiation-induced ROS damage. The authors claimed that 'Our results demonstrate a radioresistant protocell model with redox reaction system in response to ionizing radiation, which might enable the protocell fitness to prebiotic radiation on the primitive Earth preceding the emergence of enzyme-based fitness' which is acceptable. These results are an important step forwards for the artificial cell research field, also have impact in the broader scientific sense.

Overall the paper is well written. The approaches taken by the authors to build and characterise the system were clear, and most of the discussions were thorough. However, some of the conclusions drawn from the results need to be clarified. Below are some minor considerations that should be addressed, after which, I would recommend this manuscript for publication.

1. In Figure S15, it shows that droplets fusion of same type occurred rapidly. Please explain why in figure 5, the fusion only occurred between PolyP-Mn and polyP-RER-DNA droplets.
2. Also, it is not clear why polyP-RER-DNA fused into PolyP-Mn and not the other way around.
3. What was the driving force for the phase separation between the two droplets?

RESPONSE to REVIEWERS' COMMENTS

Reviewer #1 (Remarks to the Author):

The authors investigate the protection afforded by coacervates on nucleic acids and proteins. The coacervates were either made from polyphosphate and peptide (RER) or polyphosphate and Mn²⁺. The Mn²⁺ containing coacervates protected against damage from radiation whereas the peptide containing coacervates were not as effective. Surprisingly, Mn²⁺ containing coacervates did not uptake DNA but could uptake protein. Peptide containing coacervates could uptake both. This distinction is important since the authors show that only Mn²⁺ containing coacervates protect against radiation. However, since both types of coacervates could fuse with each other, the resulting structure (which the authors call a protocell) could protect DNA.

The work is unique and of clear interest to the origins field. Protecting protocells from reactive oxygen species through the activity of prebiotically plausible components that function somewhat similarly to superoxide dismutase is certainly striking. The authors used a wide variety of techniques, from spectroscopy to computational chemistry and biochemistry. I have only minor criticisms.

Reply: We appreciate the reviewer's very insightful and helpful comments. We have revised the manuscript to address the reviewer's comments as detailed below.

1. How was radiation administered? Doses of radiation are indicated, but how this was done was not clear. Clarifying this is important so that others can build on this work.

Response: Thanks to the reviewer for the comment and suggestion. The details of radiation treatment are clarified in the section of Methods (page 32 line 686-691 in the revised manuscript): "For radiation treatments, ¹³⁷Cs was used as the γ -ray radiation source (facility of the Institute of Crops and Nuclear Technology Utilization, Zhejiang Academy of Agricultural Sciences). The samples were irradiated at room temperature for 2 h with γ -ray at different dose rate. Different dose rates were achieved by adjusting the distance between samples and the γ -ray source, and doses were measured using the potassium dichromate/silver (K₂Cr₂O₇/Ag) method [*International Journal of Radiation Applications and Instrumentation. Part C. Radiation Physics and Chemistry*, 1991,38(2), 203–211. doi:10.1016/1359-0197(91)90228-t] at the assistance of the staff at the facility".

2. Did the authors try to add DNA and polyphosphate simultaneously to Mn²⁺ or change the order of addition to determine whether they could get encapsulated DNA?

Response: We appreciate the reviewer for the good point. As the reviewer suggested, we changed the order of addition to test whether the DNA could be encapsulated in the droplets. The results showed that the polyP-Mn microdroplets do not sequester DNA (as

shown in the figure below). We have included this result in the Supplementary Information (Supplementary Fig. 7). A new text is added on page 10 line 219-221: “Changing the order of addition (DNA, polyP and Mn^{2+}) had no effect on the recruitment of DNA by microdroplets, suggesting that the polyP-Mn microdroplets do not sequester DNA (Supplementary Fig. 7)”.

Supplementary Fig. 7. Effect of the order of sample addition on recruitment of DNA (FAM-ssDNA, green fluorescent) by microdroplets. Incubation conditions are as described in the Methods section. The arrow with number indicates the order of sample addition.

3. Mn^{2+} , Fe^{2+} , Ca^{2+} and Mg^{2+} were tested. Did the authors try Zn^{2+} , another metal that has been invoked in origins theories?

Response: Thanks to the reviewer for the suggestion. We have tested the polyP-Zn complexation and observed the formation of coacervate microdroplets that can efficiently sequester mCherry (as shown in the figure below). We confirmed that polyP-Zn coacervates remained intact upon exposure to 500 Gy γ -ray, while the protein mCherry lost its fluorescence after irradiation, indicating that the polyP-Zn could not effectively protect the protein under γ -ray radiation.

Figure. Microscopy images of polyP-Zn droplet and polyP-Zn droplets containing mCherry (red) following exposure to γ -ray.

4. "however, its fatty acid or phospholipid bilayers are highly impermeable to several solutes due to lack of transporters." It's fair to make this distinction with coacervates, but the permeability problem is frequently over emphasized. Fatty acid vesicles are quite permeable (doi: 10.1073/pnas.0408440102 & DOI:10.1038/nature07018) and the selective permeability of fatty acid membranes could be advantageous.

Response: We agree with the reviewer's comment. According to reviewer's comment, we have revised this sentence as "and its fatty acid or phospholipid bilayers are selectively permeable to solutes [*PNAS*, 2005;102: 6004-6008, doi: 10.1073/ pnas.0408440102; *Nature*, 2008;454, 122–125, DOI:10.1038/nature07018]."(page 3 line 50 in the revised manuscript).

5. Protocells could have emerged relatively early or late with respect to the appearance of the building blocks of life. This manuscript assumes that protocells emerged early, which is fine, but the authors should note that this is only one possibility. Perhaps a later emergence of protocells could have found milder conditions?

Response: Thanks to the reviewer for the comment and suggestion. According to the reviewer's suggestion, we have revised the related sentence to "However, there remains a missing link between prebiotic extreme scenarios such as radiations and adaptive evolution of protocells under radiation stress, if the protocells emerged early before the appearance of the building blocks of life." (page 3 line 58 in the revised manuscript). We did not exclude the possibility that a later emergence of protocells could have found milder conditions. However, at the prebiotic scenarios, the ionizing radiations including γ -ray are deleterious on biomolecules (RNA, DNA and proteins) (references 15-16), and the Mn^{2+}

containing coacervates protect against radiation and could have provided protection on DNA and proteins.

6. "The primordial intensity of terrestrial radioactivity was estimated up to 4×10^3 time higher at around 4.56 Ga than at present." This is similar to the point I was making above. People usually don't argue that life emerged at 4.56 Ga.

Response: Thanks to the reviewer for the helpful comment. According to reviewer's comment, we have removed the words "4.56 Ga" from the manuscript (page 3 line 41 and line 59 in the revised manuscript). The description on estimated primordial intensity of terrestrial radioactivity based on citation of reference 14 was used to introduce the high ionizing radiation background on the primitive Earth.

7. A comment on the prebiotic plausibility of Arg and Glu and on short peptides would be good, since RER is necessary for the described chemistry. Incidentally, tripeptide complexes with metals have been invoked before in origins scenarios (DOI: 10.1038/nchem.2817).

Response: Thanks to the reviewer for the helpful comment. The discussion and references on prebiotic plausibility of Arg, Glu and tripeptide complexes with metals are added in the revised manuscript (page 24 and 25 line 518-526): "Glutamic acids (Glu) have been identified in meteorites, which might deliver amino acids to the early Earth (*Chem Biodivers.* 2007;4(4):665-679). Peptides could be formed in the presence of minerals, which was a possible prebiotic process. It was found that Glu binds to the hydroxylapatite, a mineral of composition $\text{Ca}_5(\text{OH})(\text{PO}_4)_3$, and the basic amino acid arginine (Arg) binds to the clay minerals such as kaolinite for further polymerization by condensing agent (*Chem Biodivers.* 2007;4(4):665-679). Moreover, the tripeptide glutathione was suggested as a model prebiotic tripeptide to stabilize iron-sulfur clusters (*Nat Chem.* 2017;9 (12):1229-1234)."

8. A comment on the prebiotic concentrations of Mn^{2+} would be helpful.

Response: Thanks to the reviewer for the helpful comment. The comment on the prebiotic concentrations of Mn^{2+} is added in the revised manuscript (page 24 line 515-518): "The Mn levels in primary well-preserved carbonate rock from the Archean Eon (4 Ga to 2.5 Ga) were proposed to be up to 1% Mn (*Gondwana Research*, 2008;14:159-174). And the Mn^{2+} was inferred to have been released substantially by the high-temperature hydrothermal solutions during the Early Archean and stored in anoxic seawater in dissolved state (*Earth-Science Reviews*, 2006; 77: 273-305)."

9."Amino acids and peptides were also proved to be formed in primitive Earth conditions."
While I agree with the sentiment, "prove" is a bit too strong here.

Response: The word "prove" has been revised into "hypothesized" (page 7 line 138). We appreciate the reviewer for the helpful comment and suggestion.

Reviewer #2 (Remarks to the Author):

This manuscript demonstrated the construction of a protocell model based on coacervation of inorganic polyphosphate with divalent metal cation (Mn^{2+}) and cationic tripeptide, respectively. The biophysical properties of coacervate microdroplets has been characterized that shows the polyP-Mn droplets were radiotolerant and provides strong protection for recruited proteins, while the polyP-tripeptide droplets were not. By mixing the two populations, it was found that polyP-RER droplets fused into PolyP-Mn droplets to form polyP-RER in polyP-Mn multi-compartmentalised structure. DNA or proteins entrapped inside polyP-RER phase were efficiently protected by the polyP-Mn layer under γ -ray radiation, that was claim by nonenzymatic Mn-small molecule antioxidant complexes against radiation-induced ROS damage. The authors claimed that 'Our results demonstrate a radioresistant protocell model with redox reaction system in response to ionizing radiation, which might enable the protocell fitness to prebiotic radiation on the primitive Earth preceding the emergence of enzyme-based fitness' which is acceptable. These results are an important step forwards for the artificial cell research field, also have impact in the broader scientific sense.

Overall the paper is well written. The approaches taken by the authors to build and characterise the system were clear, and most of the discussions were thorough. However, some of the conclusions drawn from the results need to be clarified. Below are some minor considerations that should be addressed, after which, I would recommend this manuscript for publication.

Reply: We appreciate the reviewer's very insightful and helpful comments. We have revised the manuscript and addressed the reviewer's comments as detailed below.

1. In Figure S15, it shows that droplets fusion of same type occurred rapidly. Please explain why in figure 5, the fusion only occurred between PolyP-Mn and polyP-RER-DNA droplets.

Response: Thanks to the reviewer for the comment. Figure S15 (Supplementary Fig. 16 in revised Supplementary information) shows the fluidity and fusion properties of polyP

based coacervate microdroplets of same type. Actually, the fusion occurred spontaneously between the polyP based coacervate microdroplets: droplet fusion of same type droplets occurred, and the droplet fusion also occurred between the different type droplets (polyP-Mn and polyP-RER-DNA droplets). Fig. 5c and 5e shows some droplets without compartmentalized structure, which might come from the fusion of same type droplets. Here, we focused on the fusion of different type droplets, which resulted in assembly of spatially compartmentalized structure (Fig. 5b-c), while droplet fusion of same type droplets also occurred.

2. Also, it is not clear why polyP-RER-DNA fused into PolyP-Mn and not the other way around.

Response: Thanks to the reviewer for the helpful comment. The fusion of two droplets occurs to reduce the total interfacial energy. The way that two coacervate droplets fuse (i.e., 1 in 2, or 2 in 1) is largely dependent on how the interfacial energy can be minimized (*J. Am. Chem. Soc.* 2020; 142(6):2905–2914). A new discussion is now added in the revised manuscript (page 25 line 530-532).

3. What was the driving force for the phase separation between the two droplets?

Response: Thanks to the reviewer for the helpful comment. A new text is now added in the revised manuscript (page 25 line 532-536) to address this comment: "The phase separation between the two droplets is driven by complete wetting in multiphase condensate organization, which is generally due to relative interfacial tensions at play in the system containing multiple phases with different compositions (*Nature*. 2022;609 (7926):255-264; *Supramolecular Materials*, 2022;1:100019). The phase separation between the two droplets occurs to minimize the total interfacial energy (*J. Am. Chem. Soc.* 2020; 142(6):2905–2914)" .

Again, we thank the reviewers for the helpful comments and suggestions to improve the manuscript.

REVIEWERS' COMMENTS

Reviewer #1 (Remarks to the Author):

As I said previously, this is a unique and insightful study. I am happy to see this published and have no further criticisms to make, except for one.

The sentence in the introduction that reads "However, there remains a missing link between prebiotic extreme scenarios such as radiations and adaptive evolution of protocells under radiation stress, if the protocells emerged early before the appearance of the building blocks of life."

I'd suggest just ending that sentence earlier, so it reads "However, there remains a missing link between prebiotic extreme scenarios such as radiations and adaptive evolution of protocells under radiation stress, if the protocells emerged early."

Reviewer #2 (Remarks to the Author):

My concerns have been clearly addressed to my satisfaction.

RESPONSE to REVIEWERS' COMMENTS

Reviewer #1 (Remarks to the Author):

As I said previously, this is a unique and insightful study. I am happy to see this published and have no further criticisms to make, except for one.

Reply: We appreciate the reviewer's very insightful and helpful comments. We have revised the manuscript to address the reviewer's comment as detailed below.

1. The sentence in the introduction that reads "However, there remains a missing link between prebiotic extreme scenarios such as radiations and adaptive evolution of protocells under radiation stress, if the protocells emerged early before the appearance of the building blocks of life."

I'd suggest just ending that sentence earlier, so it reads "However, there remains a missing link between prebiotic extreme scenarios such as radiations and adaptive evolution of protocells under radiation stress, if the protocells emerged early."

Response: Thanks to the reviewer for the comment and suggestion. As the reviewer suggested, we have revised this sentence as "However, there remains a missing link between prebiotic extreme scenarios such as radiations and adaptive evolution of protocells under radiation stress, if the protocells emerged early." (page 3 line 66-68 in the revised manuscript).

Reviewer #2 (Remarks to the Author):

My concerns have been clearly addressed to my satisfaction.

Reply: We appreciate the reviewer's comment.

Again, we thank the reviewers for the helpful comments and suggestions to improve the manuscript.